# Risk and response adapted therapy following autologous stem cell transplant in patients with newly diagnosed multiple myeloma (RADAR (UK-MRA Myeloma XV Trial): study protocol for a phase II/III randomised controlled trial

Kara-Louise Royle ![ORCID],[1] Amy Beth Coulson ![ORCID],[1] Karthik Ramasamy,[2] David A Cairns,[1] Anna Hockaday,[1] Sergio Quezada,[3] Mark Drayson,[4] Martin Kaiser,[5] Roger Owen,[6] Holger W Auner ![ORCID],[7,8] Gordon Cook,[1,9] David Meads,[10] Catherine Olivier,[1] Lorna Barnard,[1] Rhiannon Lambkin,[1] Andrea Paterson,[1] Bryony Dawkins,[10] Mike Chapman,[11] Guy Pratt,[12] Rakesh Popat,[13] Graham Jackson,[14] Ceri Bygrave,[15] Jonathan Sive,[13] Ruth de Tute,[6] Andrew Chantry,[16] Christopher Parrish,[17] Mark Cook,[18,19] Samir Asher,[3] Kwee Yong[3]

For numbered affiliations see end of article.

**Correspondence to**
Kara-Louise Royle;
K.L.Royle@leeds.ac.uk

## ABSTRACT

**Introduction** Multiple myeloma is a plasma cell malignancy that accounts for 1%–2% of newly diagnosed cancers.

At diagnosis, approximately 20% of patients can be identified, using cytogenetics, to have inferior survival (high-risk). Additionally, standard-risk patients, with detectable disease (minimal residual disease (MRD)-positive) postautologus stem cell transplant (ASCT), fare worse compared with those who do not (MRD-negative). Research is required to determine whether a risk-adapted approach post-ASCT could further improve patient outcomes.

**Methods** RADAR is a UK, multicentre, risk-adapted, response-guided, open-label, randomised controlled trial for transplant-eligible newly diagnosed multiple myeloma patients, using combinations of lenalidomide (R), cyclophosphamide (Cy), bortezomib (Bor), dexamethasone (D) and isatuximab (Isa).

Participants receive RCyBorD(x4) induction therapy, followed by high-dose melphalan and ASCT. Post-ASCT, there are three pathways as follows:

1. A phase III discontinuation design to assess de-escalating therapy in standard-risk MRD-negative patients. Participants receive 12 cycles of Isa maintenance. Those who remain MRD-negative are randomised to either continue or stop treatment.
2. A phase II/III multiarm multistage design to test treatment strategies for treatment escalation in standard-risk MRD-positive patients. Participants are randomised to either; R, RBorD(x4) +R, RIsa, or RBorIsaD(x4) + RIsa.

3. A phase II design to assess the activity of intensive treatment strategies in high-risk patients. Participants are randomised to RBorD(x4) +R or RBorIsaD(x4) + RIsa.

1400 participants will be registered to allow for 500, 450 and 172 participants in each pathway. Randomisations are equal and treatment is given until disease progression or intolerance.

**Ethics and dissemination** Ethical approval was granted by the London–Central Research Ethics Committee (20/LO/0238) and capacity and capability confirmed by the appropriate local research and development department for each participating centre prior to opening recruitment.

## STRENGTHS AND LIMITATIONS OF THIS STUDY

⇒ RADAR trial treatment is personalised through the utilisation of baseline cytogenetic and dynamic minimal residual disease-based stratification within its design, which includes a de-escalation strategy to determine whether patients responding to treatment need to continue treatment until progression.
⇒ RADAR is adaptive, new experimental treatment arms are expected to be added as new evidence emerges.
⇒ Patients achieving <PR to induction therapy will not proceed with on-study autologus stem cell transplant and will be treated off-trial, there is no trial pathway for these participants.
⇒ RADAR excludes patients with impaired renal impairment (defined as CrCl <30 mL/min).

Participant informed consent is required before trial registration and reconfirmed post-ASCT. Results will be disseminated by conference presentations and peer-reviewed publications.

**Trial registration number** ISCRTN46841867.

## INTRODUCTION
### Multiple myeloma

Multiple myeloma (MM) is the second most common haematological malignancy with over 5500 patients diagnosed in the UK each year.[1] While there is no established cure, survival has extended markedly in the last 15 years such that patients diagnosed today can expect to live a median of 6 years, and 30% will live for 10 years or longer.[1 2]

These advances have been made possible due to the now standard use of 'novel agents' including proteasome inhibitors and immunomodulatory agents (IMiD agents) in conjunction with high-dose melphalan (HDM) supported by autologous stem cell transplantation (ASCT). While MM is predominantly a disease of older people, around 40% of patients are young and fit enough at diagnosis to be eligible for ASCT, a strategy shown in several randomised controlled trials (RCTs) to extend progression-free survival (PFS).[2–4]

Improved response rates and increased survival poses challenges for the evaluation of new regimens in a timely manner. Recent studies re-evaluating the role of ASCT in the era of novel agents have confirmed superior PFS in patients allocated to ASCT compared with those allocated to chemotherapy with deferred ASCT.[5] No overall survival (OS) benefit has been seen yet, although this probably reflects the frequent use of salvage ASCT at second remission in most trial patients. Such studies use post-ASCT continuous therapy (lenalidomide maintenance)[6] that can extend PFS and OS,[7] but may also impact on quality of life (QoL) (partly due to treatment-related toxicity), and are not without long-term risk.[8] It is thus imperative to identify those patients who are likely to benefit from extended maintenance therapy, in order to reduce healthcare costs and minimise toxicity for those who do not benefit.

### Existing evidence: induction therapy

Treatment with lenalidomide (R), cyclophosophamide (Cy), bortezomib (Bor) and dexamethasone (D) (RCyBorD) was studied as first-line therapy in the phase 2 EVOLUTION study, where it produced high response rates and deep responses compared with CyBorD or LenBorD.[9]

### Existing evidence: high-risk patients

At diagnosis, a small subgroup of patients (approximately 20%) can be identified on the basis of genetic features to have inferior survival regardless of treatment regimen (high-risk patients).[10–12] For such patients, and for those patients with refractory disease (no response to first line therapy), outcomes have remained poor despite the introduction of novel agents.[13–15] While high-risk patients often show a good response to induction therapy, the durability of the response is inferior to standard-risk patients. Using the Revised-International Staging System (R-ISS), which incorporates high-risk cytogenetic features into the the ISS, PFS time is 66 months for R-ISS stage I, 42 months for R-ISS stage II and 29 months for R-ISS stage III.[16] The attainment of minimal residual disease (MRD)-negative disease is also critical.[17–19] The activity of the RBorD regimen in patients with high-risk disease looks promising,[20] and there is a question of whether an additional benefit may be gained through the use of anti-CD38 immunotherapy. Anti-CD38[21 22] antibodies have single agent activity, but also impressive efficacy in combination, particulary with IMiD agents, lenalidomide and pomalidomide. In the RADAR trial the anti-CD38 antibody isatuximab (Isa) (SAR650984) will be used. Isatuximab is a naked Ig G1 derived monoclonal antibody (mAb) that binds selectively to a unqiue epitope on the human surface antigen called CD38.[23]

### Existing evidence: standard-risk patients

With increasingly effective multidrug regimens being used for induction treatment, and responses further deepened following high-dose therapy and ASCT, the major challenge for standard-risk participants is how to maintain their disease control for as long as possible. The benefit of post-ASCT consolidation approaches remains controversial, with inconsistent evidence identifying both appropriate patient groups and treatment strategies.

Maintenance lenalidomide (R) has been studied within several large studies and has consistently been shown to increase PFS, confmed within a recent meta-analysis.[7] The impact of long-term lenalidomide therapy on bone marrow and immune function, and general well-being are not completely understood. Thus there is an urgent need to identify those patients who benefit most from this approach, as well as an evaluation of other agents that use the host immune system to improve outcomes, such as antibodies. It is also of benefit to explore the option and effect of stopping maintenance therapy in patients who have achieved and maintained the deepest response

### Existing evidence: use of MRD for stratifying treatment in standard-risk patients

MRD status is a clinically relevant biomarker, and an emerging clinical trial endpoint with recent guidance published by the International Myeloma Working Group (IMWG).[19 24]

Several studies have shown that patients who have detectable disease in their bone marrow (MRD-positive) following ASCT suffer earlier relapses and shorter survival when compared with those who are MRD-negative.[25 26] These results have been combined in a recent meta-analysis.[26] Hence these patients should be the focus of investigative strategies aimed at deepening disease response and converting them from MRD-positive to MRD-negative.

Patients without detectable disease in their bone marrow (MRD-negative), have longer PFS and OS .[27] In the Intergroupe Francophone du Myelome (IFM) 2009 study that randomised patients to receive ASCT upfront or to continue chemotherapy, patients who were MRD-negative fared equally well, regardless of whether they had received ASCT, or chemotherapy alone. This would suggest that those who have achieved MRD-negative response may not need to undergo intensive consolidation therapy, and instead may benefit from a de-escalation of therapy.

## Aims and objectives

RADAR aims to address the most important contemporary therapeutic questions for newly diagnosed MM transplant eligible (NDMM TE) patients post-ASCT by considering three different post-ASCT treatment pathways:

1. A de-escalation treatment strategy is used to determine whether ceasing Isa maintenance treatment, after 12 cycles of continuous treatment, is non-inferior to long-term Isa treatment, in terms of PFS in participants who are standard-risk at diagnosis (participants who have ≤1 high-risk cytogenetic lesion) and are MRD-negative at 100 days post-ASCT and 12 months (cycles) after.

2. A treatment escalation strategy is used for participants who are standard-risk at diagnosis (participants who have ≤1 high-risk cytogenetic lesion) and MRD-positive at 100 days post ASCT. The trial aims to compare the activity and efficacy of post-ASCT consolidation and maintenance treatment between combinations of R, Bor D and Isa to R alone in terms of conversion to MRD-negativity.

3. Dose-intensive consolidation and maintenance strategies will be studied for participants who are high-risk at diagnosis (participants who have >1 high-risk cytogenetic lesion). The activity of post-ASCT strategies containing R, Bor, D and Isa will be assessed in terms of PFS rate.

## Trial design

RADAR is a national, multicentre, risk-adapted, response-guided open-label, RCT for NDMM TE patients: phase II/III for standard-risk, MRD-Positive participants, phase III randomised discontinuation design for standard-risk, MRD-Negative participants, phase II randomised for high-risk participants. The following report details the trial protocol and follows the structure of the Standard Protocol Items: Recommendations for Interventional Trials (SPIRIT) statement.[28] The SPIRIT checklist can be found within online supplemental material 1.

## METHODS
## Setting

The RADAR trial will be conducted at 70 centres in the UK (see online supplemental material 1), as identified via a feasibility assessment to determine those most appropriate to participate in the trial.

Potential participants will be identified by the research team at the time they are referred to the haematology outpatient department with suspected MM. A smaller number of participants may be identified during inpatient admissions. Invitation to participate in the trial and provision of information will be made either during their first consultation, when routine diagnostic tests will be performed and potential treatment options discussed, or at the time they receive their diagnostic test results.

## Eligibility criteria

Adults (18 years and older) with previously untreated MM requiring therapy, as defined by IMWG diagnostic criteria, who are eligible for stem cell transplant and capable of giving informed consent will be assessed for eligibility. Eligibility will be confirmed prior to; registration, the start of isatuximab maintenance treatment in the Randomisation 1 pathway and each randomisation by the principal investigator (PI) or an appropriately authorised medically trained delegate and will be recorded in the participants' medical records and on the relevant electronic case report form (eCRF).

To be eligible for each point in the trial the participants must meet all the inclusion criteria and none of the exclusion criteria presented in box 1 for trial registration, box 2 for isatuximab treatment in the R1 pathway, box 3 for Randomisation 1 (R1), box 4 for Randomisation 2 (R2), box 5 for Randomisation 3 (R3).

## Interventions and dosing
### Intervention schedule

All participants will receive 4, 21 days, cycles of induction treatment with RCyBorD. During this time, the genetics risk status of the participant will be analysed and determined from bone marrow samples prior induction treatment (from diagnosis or trial-specific procedures). Participants will either be allocated as high-risk (defined as the presence of at least two of these adverse lesions: t(4,14), t(14,16), t(14,20), del(17p) or gain(1q)) or standard risk. At the end of the induction cycle, participants will be assessed for response In the absence of disease progression or intolerance, participants with at least a partial response (PR) to will proceed to HDM and ASCT, which will be carried out according to local practice. At 100 days post-ASCT participants will be assessed for MRD and response and will follow one of three pathways of post ASCT treatment, provided they meet the eligibility criteria:

1. Participants identified as standard-risk and who are MRD-negative post-ASCT will be treated according to the R1 pathway. All participants will receive 12 cycles of Isa maintenance treatment. Those who remain MRD-negative following cycle 12, will be randomised to continue or stop Isa maintenance treatment.

2. Participants identified as standard-risk and who are MRD-positive post-ASCT and have at least a minimal response (MR) at 100 days post ASCT, will proceed to R2. Eligible participants will be randomised between:

## Box 1  Inclusion and exclusion criteria for main trial registration

### Inclusion criteria

1. Previously untreated patients with multiple myeloma (MM) requiring therapy, defined as having myeloma defining events or with biomarkers of malignancy according to International Myeloma Working Group diagnostic criteria.
2. Eligible for stem cell transplant.
3. Eastern Cooperative Oncology Group (ECOG) performance status 0–2 (except in cases where ECOG >2 is due to effects of myeloma eg, spinal cord compression);
4. Total bilirubin ≤1.5x upper limit of normal (ULN).
5. Alanine aminotransferase (ALT) and/or aspartate aminotransferase (AST) ≤3x ULN (if ALT and AST are tested, both must meet this criteria).
6. Adequate marrow function:
   - Neutrophils ≥1.0 × $10^9$/L (unless the participant has a known/suspected diagnosis of familial or racial neutropenia in which case an Absolute neutrophil count (ANC) ≥0.75x $10^9$/L is allowed).
   - Haemaglobin ≥80 g/L. Blood transfusions within 3 days prior to eligibility assessments are not permitted.
   - Platelets ≥75 × $10^9$/L (in the case of heavy bone marrow infiltration (>50%) which is, in the opinion of the investigator, the cause of the thrombocytopaenia and provided appropriate supportive measures and patient monitoring are in place, a platelet count of ≥50 × $10^9$/L is permitted. Platelet transfusions within 3 days prior to eligibility assessments are not permitted.
7. Creatinine clearance ≥30 mL/minute, according to the Cockcroft-Gault formula, following correction of reversible causes (eg, dehydration, hypercalcaemia, sepsis).
8. Able to swallow oral medication.
9. Aged at least 18 years.
10. Agree to follow the pregnancy prevention guidelines:
    Female participants who:
    a. Are not of childbearing potential, OR
    b. If they are of childbearing potential, agree to practise two effective methods of contraception, at the same time, from the time of signing the informed consent form until 12 months after the last dose of cyclophosphamide or 5 months after the last dose of study drug, whichever is longest.
    c. Agree to practise true abstinence when this is in line with the preferred and usual lifestyle of the subject from the time of signing the informed consent form until 12 months after the last dose of cyclophosphamide or 5 months after the last dose of study drug, whichever is longest. (Periodic abstinence (eg, calendar ovulation, symptothermal, postovulation methods) and withdrawal are not acceptable methods of contraception.)
    d. Agree to not donate oocytes during the entire study treatment period and until 12 months after the last dose of cyclophosphamide or 5 months after the last dose of study drug, whichever is longest.
    Male participants, even if surgically sterilised (ie, status post-vasectomy), must agree to one of the following:
    a. Agree to practise effective barrier contraception during the entire study treatment period and until 6 months after the last dose of cyclophosphamide or 5 months after the last dose of study drug, whichever is longest.
    b. Agree to practise true abstinence when this is in line with the preferred and usual lifestyle of the subject, during the entire study treatment period and until 6 months after the last dose of

*Continued*

## Box 1   Continued

cyclophosphamide or 5 months after the last dose of study drug, whichever is longest. (Periodic abstinence (eg, calendar ovulation, symptothermal, postovulation methods) and withdrawal are not acceptable methods of contraception).

c. Agree to not donate sperm during the entire study treatment period and until 6 months after the last dose of cyclophosphamide or 5 months after the last dose of study drug, whichever is longest.

Contraception for female and male participants must be in accordance with (and participants must consent to) the Celgene Pregnancy Prevention Plan.

If female and of childbearing potential, they must have a negative pregnancy test performed by a healthcare professional within 14 days prior to registration.

11. Able to provide written informed consent.

### Exclusion criteria

1. Asymptomatic (smouldering) MM, monoclonal gammopathy of undetermined significance, solitary plasmacytoma of bone or extramedullary plasmacytoma (without evidence of MM).
2. Received previous treatment for MM, with the exception of local radiotherapy to relieve bone pain or spinal cord compression, prior bisphosphonate treatment, or corticosteroids as long as the total dose does not exceed the equivalent of 160mg dexamethasone. This criteria is not applicable at R1, R2 and R3 when participants will have received previous treatment for MM as part of this trial.
3. Unstable angina or myocardial infarction within 4 months prior to registration (or at any time since registration for participants starting isatuximab maintenance, R1, R2 and R3), New York Heart Association (NYHA) class III or IV heart failure, uncontrolled angina, history of severe coronary artery disease, severe uncontrolled ventricular arrhythmias, sick sinus syndrome, or electrocardiographic evidence of acute ischaemia or grade 3 conduction system abnormalities unless subject has a pacemaker.
4. Cardiac disorder identified according to local practice (eg, left ventricular ejection fraction; results from formal measurements acceptable within 28 days prior to registration).
5. Significant neuropathy (grade ≥3 or grade 2 with pain).
6. Prior malignancy that required treatment or has shown evidence of recurrence (except for non-melanoma skin cancer or adequately treated cervical carcinoma in situ) during the 5 years prior to registration. Cancer treated with curative intent for >5 years previously and without evidence of recurrence will be allowed.
7. Pregnant, lactating or breastfeeding female participants.
8. Known resistance, intolerance or hypersensitivity to any component of the planned therapies, except in the case of hypersensitivity which is amenable to premedication with steroids or H2 blocker. Intolerance includes hereditary problems of galactose intolerance, the Lapp lactase deficiency or glucose-galactose malabsorption.
9. Major surgery within 14 days before registration (or starting isatuximab maintenance, R1, R2 and R3). This would include surgical intervention for relief of cord compression but does not include vertebroplasty or kyphoplasty.
10. Known gastrointestinal (GI) disease or GI procedure that could interfere with the oral absorption or tolerance of trial treatment, including difficulty swallowing.
11. Active systemic infection.
12. Participant is hepatitis B surface antigen positive, hepatitis C antibody positive or HIV positive (participants who are hepatitis C core antibody positive but have been successfully treated for the

*Continued*

## Box 1    Continued

disease may still be eligible—please consult the Clinical Trials Research Unit/CI. Participants must have hepatitis and HIV screening conducted within 28 days prior to registration.

13. Any other medical or psychiatric condition which, in the opinion of the investigator, contraindicates the participant's participation in this study.

14. Receipt of live vaccination within 30 days prior to registration, for the duration of the study and for 3 months after the last dose of study drug.

15. Participant has risk factors for thromboembolism including the use of agents which may increase their risk of thrombosis, such as hormone replacement therapy (this exclusion criteria is applicable only at registration and when starting R2 or R3 pathway).

16. Participant has risk factors for seizures (this exclusion criteria is applicable only at registration and when starting R2 or R3 pathway).

R maintenance; RBorD consolidation and R maintenance (RBorD+R); RISa maintenance; or RBorIsaD consolidation and RIsa maintenance (RBorIsaD+RIsa).

3. Participants identified as high-risk and have at least a PR at 100 days post-ASCT will proceed to R3. These participants will be randomised to receive either RBorD consolidation, followed by R maintenance; or RBorIsaD consolidation followed by RIsa maintenance.

All randomisations are balanced across the treatment arms. A computer-generated minimisation programme that incorporates a random element will be used to ensure treatment groups are well balanced for the stratification factors of each randomisation (see table 1).

Table 2 summarises the starting doses for the induction treatment and each post-ASCT treatment pathway. Following ASCT all treatments will continue until disease progression or intolerance. Response will be assessed at the end of each cycle according to the IMWG 2016 response criteria.[24 29–31] Treatment intolerance will be assessed throughout each treatment cycle, according to the National Cancer Institute (NCI) Common Terminology Criteria for Adverse Events (CTCAE) V.5. All consolidation cycles are 21 days in length, maintenance cycles are 28 days in length.

### Intervention adherence

Throughout the trial R will be taken orally and swallowed whole at the same time on the scheduled days. D and Cy will be taken orally, while Bor will be administrated via a subcutaneous injection. Bortezomib may also be administered at home if this is in line with local policies and if appropriate procedures are in place. Isa will be given as an intravenous infusion and at a facility capable of managing hypersensitivity reactions.

To monitor adherence to the oral trial medicines, participants will complete a participant diary card which will be reviewed at each trial visit. Non-compliance will be reported to the Clinical Trials Research Unit (CTRU) and any unused capsules will be returned to the pharmacy.

## Box 2    Inclusion and exclusion criteria for isatuximab treatment (R1 pathway)

### Inclusion criteria

1. Standard-risk (participant is **not confirmed to have** at least two of these genetically adverse lesions: t(4;14), t(14;16), t(14;20), del(17p), gain(1q)), as confirmed by the Clinical Trials Research Unit.

2. Four cycles of RCyBorD received.

3. Minimal residual disease (MRD)-negative (proportion of malignant cells in the bone marrow is <1 in 100 000, confirmed by Haematology Malignancy Diagnostic Service (HMDS) central lab) at 100 days postautologus stem cell transplant (ASCT).

4. Received ≥100 mg/m$^2$ high-dose melphalan and ASCT.

5. Eastern Cooperative Oncology Group (ECOG) performance status 0–2 (except in cases where ECOG>2 is due to effects of myeloma, eg, spinal cord compression).

6. Total bilirubin ≤1.5×upper limit of normal (ULN).

7. Alanine aminotransferase (ALT) and/or aspartate aminotransferase (AST)≤3×ULN (if ALT and AST are tested, both must meet this criteria).

8. Adequate marrow function:
   a. neutrophils ≥1.0 × 10$^9$/L (unless the participant has a known/suspected diagnosis of familial or racial neutropenia in which case an ANC ≥0.75×10$^9$/L is allowed).
   b. Hb ≥80 g/L. Blood transfusions within 3 days prior to eligibility assessments are not permitted.
   c. Platelets ≥75 × 10$^9$/L. Platelet transfusions within 3 days prior to eligibility assessments are not permitted.

9. Creatinine clearance ≥30 mL/minute, according to the Cockcroft-Gault formula, following correction of reversible causes (eg, dehydration, hypercalcaemia, sepsis).

10. Agree to follow the pregnancy prevention guidelines:
    Female participants who:
    a. Are not of childbearing potential.
    b. If they are of childbearing potential, agree to practise two effective methods of contraception, at the same time, from the time of signing the informed consent form until 12 months after the last dose of cyclophosphamide or until 5 months after the last dose of study drug, whichever is longest.
    c. Agree to practise true abstinence when this is in line with the preferred and usual lifestyle of the subject, from the time of signing the informed consent form until 12 months after the last dose of cyclophosphamide or until 5 months after the last dose of study drug, whichever is longest. (Periodic abstinence (eg, calendar, ovulation, symptothermal, post-ovulation methods) and withdrawal are not acceptable methods of contraception.)
    d. Agree to not donate oocytes during the entire study treatment period until 12 months after the last dose of cyclophosphamide or 5 months after the last dose of study drug, whichever is longest.
    Male participants, even if surgically sterilised (ie, status postvasectomy), must agree to one of the following:
    a. Agree to practise effective barrier contraception during the entire study treatment period and until 6 months after the last dose of cyclophosphamide or until 5 months after the last dose of study drug, whichever is longest.
    b. Agree to practise true abstinence when this is in line with the preferred and usual lifestyle of the subject during the entire study treatment period and until 6 months after the last dose of cyclophosphamide or until 5 months after the last dose of study drug, whichever is longest. (Periodic abstinence (eg, calendar,

Continued

## Box 2   Continued

ovulation, symptothermal, postovulation methods) and withdrawal are not acceptable methods of contraception).

c.  Agree to not donate sperm during the entire study treatment period until 6 months after the last dose of cyclophosphamide or 5 months after the last dose of study drug, whichever is longest.

11.  Signed the informed consent document for the R1 treatment pathway.

### Exclusion criteria

1.  Disease progression (according to International Myeloma Working Group criteria).
2.  MRD-positive (proportion of malignant cells in the bone marrow is ≥1 in 100 000, confirmed by HMDS central lab) at 100 days post-ASCT.
3.  Registration exclusion criteria (cardiac risk assessment, HIV and Hepatitis B and C testing do not need to be repeated).

### Dose modifications and discontinuations

With the exception of Isa, which can only be interrupted and resumed, dose modifications are permitted in the management of toxicities. Upfront dose modifications in response to liver and/or renal impairment are also permitted within the protocol. When the dose of any drug is reduced the dose cannot be re-escalated in this trial.

If one drug in a combination is stopped, treatment with the other drugs in combination can continue.

Scenarios in which the participant may remain on trial following consultation with the chief investigator (CI) are if they: discontinue two or more drugs during combination therapy; cease treatment for more than 3 weeks in between any treatment cycle or are delayed in starting maintenance of consolidation beyond 120 days post-ASCT. Note that if Bor, R or D is stopped during the pre-ASCT induction treatment due to toxicity, the participant will only be able to be treated on the standard-risk-MRD-negative pathway (assuming they meet the relevant inclusion criteria).

If treatment is discontinued early, the participant will be treated off trial at the discretion of their treating clinician.

### Concomitant medication

Local support care protocols, including antiemetic schedules, tumour lysis syndrome prevention, venous thromboembolism prophylaxis and prophylactic antimicrobial therapy should be followed. The excluded concomitant medications and procedures while receiving trial treatment are listed in online supplemental material.

Concomitant medication, disease and other malignancies will be recorded at eligibility.

### Outcomes
#### Primary outcomes

PFS-R1 is defined as the time from R1 to the time of first documented evidence of disease progression or death from any cause. Individuals who are lost to follow-up or progression-free at the time of analysis will be censored at their last known date to be alive and progression-free.

## Box 3   Inclusion and exclusion criteria for randomisation 1

### Inclusion criteria

1.  Standard-risk (participant is **not confirmed to have** at least two of these genetically adverse lesions: t(4;14), t(14;16), t(14;20), del(17p), gain(1q)), as confirmed by the Clinical Trials Research Unit.
2.  Twelve cycles of isatuximab maintenance received.
3.  Minimal residual disease (MRD)-negative (proportion of malignant cells in the bone marrow is <1 in 100 000, confirmed by HMDS central lab) at 100 days postautologus stem cell transplant (ASCT).
4.  MRD-negative (proportion of malignant cells in the bone marrow is <1 in 100,000, confirmed by HMDS central lab) after 12 cycles of isatuximab.
5.  Received ≥100 mg/m$^2$ high-dose melphalan and ASCT.
6.  Eastern Cooperative Oncology Group (ECOG) performance status 0–2 (except in cases where ECOG >2 is due to effects of myeloma, eg, spinal cord compression).
7.  Total bilirubin ≤1.5 × upper limit of normal (ULN).
8.  Alanine aminotransferase (ALT) and/or aspartate aminotransferase (AST)≤3 x ULN (if ALT and AST are tested, both must meet this criteria).
9.  Adequate marrow function:
    a.  Neutrophils ≥1.0 × 10$^9$/L (unless the participant has a known/suspected diagnosis of familial or racial neutropenia in which case an ANC ≥0.75 . 10$^9$/L is allowed),
    b.  Hb≥80 g/L. Blood transfusions within 3 days days prior to eligibility assessments are not permitted.
    c.  Platelets ≥75 × 10$^9$/L . Platelet transfusions within 3 days days prior to eligibility assessments are not permitted.
10.  Creatinine clearance ≥30 mL/minute, according to the Cockcroft-Gault formula, following correction of reversible causes (eg, dehydration, hypercalcaemia, sepsis).
11.  Agree to follow the pregnancy prevention guidelines:
12.  Female participants who:
    a.  Are not of childbearing potential.
    b.  If they are of childbearing potential, agree to practise two effective methods of contraception, at the same time, from the time of signing the informed consent form until 5 months after the last dose of study drug.
    c.  Agree to practise true abstinence when this is in line with the preferred and usual lifestyle of the subject from the time of signing the informed consent form until 5 months after the last dose of study drug. (Periodic abstinence (eg, calendar, ovulation, symptothermal, postovulation methods) and withdrawal are not acceptable methods of contraception.)
    d.  Agree to not donate oocytes during the entire study treatment period and until 5 months after the last dose of study drug.
13.  Male participants, even if surgically sterilised (ie, status postvasectomy), must agree to one of the following:
    a.  Agree to practise effective barrier contraception during the entire study treatment period and until 5 months after the last dose of study drug.
    b.  Agree to practise true abstinence when this is in line with the preferred and usual lifestyle of the subject during the entire study treatment period and until 5 months after the last dose of study drug. (Periodic abstinence (eg, calendar, ovulation, symptothermal, postovulation methods) and withdrawal are not acceptable methods of contraception.

Continued

## Box 3  Continued

 c. Agree to not donate sperm during the entire study treatment period and until 5 months after the last dose of study drug.
14. Signed the informed consent document for the R1 treatment pathway.

### Exclusion criteria
1. Disease progression (according to International Myeloma Working Group criteria).
2. MRD-positive (proportion of malignant cells in the bone marrow is ≥1 in 100 000, confirmed by HMDS central lab) at 100 days post-ASCT or after 12 cycles of isatuximab.
3. Registration exclusion criteria (cardiac risk assessment, HIV and hepatitis B and C testing do not need to be repeated).

Attainment of MRD-negativity is defined as a binary endpoint. MRD-negativity will be determined at 6 months post-R2 (end of cycle 6 post-ASCT treatment for participants allocated to maintenance only strategies and end of cycle 7 post-ASCT treatment for participants allocated to maintenance and consolidation strategies) according to the IMWG 2016 response criteria.

The PFS rate is defined as the proportion of participants who are alive and progression-free 28 months post-R3.

Disease progression is defined according to the IMWG 2016 response criteria for MM.[24 29–31]

### Secondary outcomes
The secondary outcomes of this trial are to assess; PFS for R2 and R3, Time to progression, Time to second PFS event (PFS2), OS, Survival after progression, Objective response rate, Attainment of Very Good PR, Attainment of MRD negativity, Duration of MRD Negativity, Time to improved response, Time to next treatment, Treatment compliance and total amount of therapy delivered, Toxicity and Safety, including the incidence of second malignancies, QoL including EORTC-QLQ-C30 (European Organisation for Research and Treatment of Cancer Core Quality of Life questionnaire), EORTC-QLQ-MY20 (EORTC Myeloma Module) and EQ-5D-3L (EuroQol-5Dimension-3Level) and cost-effectiveness.

### Exploratory and subgroup analyses
An overview of the planned exploratory and subgroup analysis can be found in online supplemental material. These include genetic and molecular analysis of participant samples conducted by the respective central laboratories, as well as additional analysis to review the infection and re-infection rates of COVID-19 in participants during induction, post-ASCT and during post-ASCT consolidation.

### Participant's timelines
The full trial schema can be seen in figure 1. The schedule of local assessments at each time point are presented in figure 2.

## Box 4  Inclusion and exclusion criteria for randomisation 2

### Inclusion criteria
1. Standard-risk (participant is **not confirmed to have** at least two of these genetically adverse lesions: t(4;14), t(14;16), t(14;20), del(17p), gain(1q)) as confirmed by Clinical Trials Research Unit.
2. Four cycles of RCyBorD received.
3. Minimal residual disease-positive (proportion of malignant cells in the bone marrow is ≥1 in 100 000, confirmed by HMDS central lab) at 100 days postautologus stem cell transplant (ASCT).
4. Received ≥100 mg/m$^2$ high-dose melphalan and ASCT.
5. Eastern Cooperative Oncology Group (ECOG) performance status 0–2 (except in cases where ECOG>2 is due to effects of myeloma, eg, spinal cord compression).
6. Total bilirubin ≤1.5× upper limit of normal (ULN).
7. Alanine aminotransferase (ALT) and/or aspartate aminotransferase (AST) ≤3× ULN (if ALT and AST are tested, both must meet these criteria).
8. Adequate marrow function:
 a. Neutrophils≥1.0 × 10$^9$/L(unless the participant has a known/suspected diagnosis of familial or racial neutropenia in which case an ANC ≥0.75× 10$^9$/L is allowed).
 b. Hb≥80 g/L.
 c. Platelets ≥75 × 10$^9$/L. Platelet transfusions within 3 days days prior to eligibility assessments are not permitted.
9. Creatinine clearance ≥30 mL/minute, according to the Cockcroft-Gault formula, following correction of reversible causes (eg, dehydration, hypercalcaemia, sepsis).
10. Agree to follow the pregnancy prevention guidelines:
 Female participants who:
 a. Are not of childbearing potential.
 b. If they are of childbearing potential, agree to practise two effective methods of contraception, at the same time, from the time of signing the informed consent form until 12 months after the last dose of cyclophosphamide or 5 months after the last dose of study drug, whichever is longest.
 c. Agree to practise true abstinence when this is in line with the preferred and usual lifestyle of the subject, from the time of signing the informed consent form until 12 months after the last dose of cyclophosphamide or 5 months after the last dose of study drug, whichever is longest. (Periodic abstinence (eg, calendar, ovulation, symptothermal, postovulation methods) and withdrawal are not acceptable methods of contraception).
 d. Agree to not donate oocytes during the entire study treatment period and until 12 months after the last dose of cyclophosphamide or 5 months after the last dose of study drug, whichever is longest.
 Male participants, even if surgically sterilised (ie, status post-vasectomy), must agree to one of the following:
 a. Agree to practise effective barrier contraception during the entire study treatment period and until 6 months after the last dose of cyclophosphamide or 5 months after the last dose of study drug, whichever is longest.
 b. Agree to practise true abstinence when this is in line with the preferred and usual lifestyle of the subject during the entire study treatment period and until 6 months after the last dose of cyclophosphamide or 5 months after the last dose of study drug, whichever is longest. (Periodic abstinence (eg, calendar, ovulation, symptothermal, postovulation methods) and withdrawal are not acceptable methods of contraception).

Continued

## Box 4 Continued

 c. Agree to not donate sperm during the entire study treatment period and until until 6 months after the last dose of cyclophosphamide or 5 months after the last dose of study drug, whichever is longest.

11. Signed the informed consent document for the R2 treatment pathway.

### Exclusion criteria:

1. Disease progression (according to International Myeloma Working Group criteria).
2. Registration exclusion criteria (cardiac risk assessment, HIV and Hepatitis B and C testing do not need to be repeated).

### Trial entry

Participants will enter into the trial at one of two points in their pathway, this will either be at bone marrow registration or at main trial registration.

### Trial consent

Participants who enter the trial at bone marrow registration will provide consent to have bone marrow samples taken and sent to central laboratories for analysis. If the participant does not have myeloma or decides to not take part in RADAR, they will have the option of consenting to their samples being used in future research.

All participants will be required to provide written informed consent for the trial prior to trial registration (for initial trial treatment and further trial information) and prior to starting their post-ASCT treatment pathway once their post-ASCT assessments have been completed (for their pathway specific treatment). Optional consent regarding QoL and healthcare resource use questionnaires and the use of samples for future research will also be obtained within the first consent process.

Consent forms are included in online supplemental material.

### Trial registration

Following initial consent, participant eligibility for trial registration will be assessed. Trial samples for blood and urine will be taken for all participants and bone marrow samples will be taken for those who did not enter the trial through bone marrow registration.

### Genetic risk

Bone marrow samples from diagnosis will be analysed at time of diagnosis at local cytogenetics laboratories for FISH (Fluorescence in-situ hybridisation)-defined high-risk makers. Original/copy of the FISH testing report will also be sent to CTRU where they will be reviewed by the CI/Trial management team for risk stratification. Participants will either be categorised high-risk (defined as the presence of at least 2 of the following adverse lesions: t(4;14), t(4;16), t(14;20), del(17p) or gain(1q)) or standard-risk.

Participants will commence RCyBorD treatment once a diagnosis of symptomatic myeloma has been confirmed

## Box 5 Inclusion and exclusion criteria for randomisation 3

### Inclusion criteria

1. High-risk (participant **is confirmed to have** at least two of these genetically adverse lesions: t(4;14), t(14;16), t(14;20), del(17p), gain(1q)) as confirmed by Clinical Trials Research Unit.
2. Four cycles of RCyBord received.
3. Received ≥100 mg/m$^2$ high-dose melphalan and autologus stem cell transplant.
4. Eastern Cooperative Oncology Group (ECOG) performance status 0–2 (except in cases where ECOG>2 is due to effects of myeloma, eg, spinal cord compression).
5. Total bilirubin ≤1.5 × upper limit of normal (ULN)
6. Alanine aminotransferase (ALT) and/or aspartate aminotransferase (AST)≤3 x ULN (if ALT and AST are tested, both must meet this criteria).
7. Adequate marrow function:
   a. Neutrophils≥1.0 × 10$^9$/L (unless the participant has a known/suspected diagnosis of familial or racial neutropenia in which case an ANC ≥0.75 × 109/L is allowed).
   b. Hb≥80 g/L.
   c. Platelets ≥75 × 10$^9$/L. Platelet transfusions within 3 days days prior to eligibility assessments are not permitted.
   d. Creatinine clearance (CrCl) ≥30 mL/minute, according to the Cockcroft-Gault formula, following correction of reversible causes (eg, dehydration, hypercalcaemia, sepsis).
8. CrCl≥30 mL/minute, according to the Cockcroft-Gault formula, following correction of reversible causes (eg, dehydration, hypercalcaemia, sepsis).
9. Agree to follow the pregnancy prevention guidelines:
   Female participants who:
   a. Are not of childbearing potential.
   b. If they are of childbearing potential, agree to practise two effective methods of contraception, at the same time, from the time of signing the informed consent form until 12 months after the last dose of cyclophosphamide or 5 months after the last dose of study drug, whichever is longest.
   c. Agree to practise true abstinence when this is in line with the preferred and usual lifestyle of the subject from the time of signing the informed consent form until 12 months after the last dose of cyclophosphamide or 5 months after the last dose of study drug, whichever is longest. (Periodic abstinence (eg, calendar, ovulation, symptothermal, postovulation methods) and withdrawal are not acceptable methods of contraception).
   d. Agree to not donate oocytes during the entire study treatment period and until 12 months after the last dose of cyclophosphamide or 5 months after the last dose of study drug, whichever is longest.
   Male participants, even if surgically sterilised (ie, status postvasectomy), must agree to one of the following:
   a. Agree to practise effective barrier contraception during the entire study treatment period and until 6 months after the last dose of cyclophosphamide or 5 months after the last dose of study drug, whichever is longest.
   b. Agree to practise true abstinence when this is in line with the preferred and usual lifestyle of the subject during the entire study treatment period and until 6 months after the last dose of cyclophosphamide or 5 months after the last dose of study drug, whichever is longest. (Periodic abstinence (eg, calendar,

Continued

## Box 5   Continued

ovulation, symptothermal, postovulation methods) and withdrawal are not acceptable methods of contraception).
  c. Agree to not donate sperm during the entire study treatment period and until 6 months after the last dose of cyclophosphamide or 5 months after the last dose of study drug, whichever is longest.
10. Signed the informed consent document for the R3 treatment pathway.

### Exclusion criteria
1. Disease progression (according to International Myeloma Working Group criteria).
2. Registration exclusion criteria (cardiac risk assessment, HIV and hepatitis B and C testing do not need to be repeated).

by the treating physician, and ideally when it is known that the FISH testing has been successful.

Where risk status can be determined from the first sample participants can continue on the trial. If status cannot be confirmed and the FISH report shows a full test failure that is, no information relating to adverse lesions is available, a second attempt will be made to collect a bone marrow sample prior to treatment starting. If partial results are obtained from the FISH sample, it may be possible that the CI/ trial management group (TMG) will be able to determine risk status using the assumption that translocations are mutually exclusive and that some markers are rare. If risk status can be determined a second bone marrow sample is not required. However, this must be confirmed by the CTRU.

If, after two bone marrow samples, the FISH results for some or all the adverse lesions are inconclusive participants risk will be determined using a combination of the results from the two samples. The most appropriate pathway will be determined by the CI. Note this may result in the participants stratification factor being 'unable to determine' at randomisation. If a second bone marrow has been requested by CTRU but the collection was not attempted, the participants will be taken off the study (after ASCT, if induction treatment was already started).

### Induction and ASCT
Following trial registration, once all central samples have been taken and risk status confirmed, participants will receive induction treatment as described in the intervention section. In the absence of disease progression or treatment intolerance participants with at least a PR to will proceed to HDM and ASCT.

Consenting participants will complete QoL and resource use questionnaires at the end of RCyBorD induction treatment.

All participants who go through ASCT will have a bone marrow sample taken at 100 days (±14) post-ASCT. If an MRD result cannot be obtained at the first attempt, the bone marrow will be repeated once (within 28 days after 100 days post-ASCT time point). If the second attempt fails, the participant will be considered MRD-positive.

### Post-ASCT treatment
Participants who receive all 4 cycles of induction treatment followed by ASCT will be assessed for eligibility into their assigned randomisation pathways determined by genetic risk and MRD results and treated as described in the intervention schedule.

Consenting participants in the R1 pathway will complete QoL and resource use questionnaires after cycle 6, 12, 24 and 36 (participants randomised to 'stop Isa' will complete the cycle 24 and 36 QoL and resource use questionnaires at equivalent time points).

Similarly, consenting participants to receive R or RIsa within pathway R2 and R3 will complete QoL and resource use questionnaires after cycles 6, 12 and 24 of maintenance. Those randomised to receive RBorIsa+R or RBorIsaD+RIsa will complete QoL and resource use questionnaires after cycles 3, 9 and 21 of maintenance.

Post-ASCT treatment (including observation only in the R1 pathway) is continued until disease progression, participant or clinician withdrawal or death. After treatment discontinuation, the participant is treated off trial by their local clinician and followed-up as per the trial follow-up schedule.

| Table 1 | Stratification factors |
|---|---|
| **Randomisation:** | **Stratification factors:** |
| 1 | ► Centre<br>► No of high risk lesions at trial registration (0 or 1 or unable to determine) |
| 2 | ► Centre<br>► No of high-risk lesions at trial registration (0 or 1 or unable to determine)<br>► Response at 100 days post-ASCT (< *VGPR*, ≥ *VGPR*) |
| 3 | ► Centre<br>► No of high-risk lesions at trial registration (2, or ≥3, or unable to determine)<br>► Response at 100 days post-ASCT (< *VGPR*, ≥ *VGPR*) |

ASCT, autologus stem cell transplant; VGPR, Very Good Partial Response.

**Table 2** Dosing schedule for RCyBorD induction and each randomisation pathway

**Trial registration—induction (each cycle = 21 days)**

| Treatment (route) | Induction—all participants (starting dose) | Days |
|---|---|---|
| Bortezomib (SC) | 1.3 mg/m$^2$ | 1, 8, 15 |
| Cyclophosphamide (PO) | 500 mg | 1, 8 |
| Lenalidomide (PO) | 25 mg | 1–14 |
| Dexamethasone (PO) | 40 mg | 1, 8, 15 |

**Isatuximab pre-randomisation 1 (each cycle = 28 days)**

| Treatment (route) | R1 pathway participants only (starting dose) | Days |
|---|---|---|
| Isatuximab (IV) | 10 mg/kg | 1, 8, 15 and 22 cycle 1 then days 1 and 15 from cycle two onwards |

**Randomisation 1: Isatuximab maintenance (each cycle = 28 days)**

| Treatment (route) | Isatuximab (days) | "Stop Isatuximab" |
|---|---|---|
| Isatuximab (IV) | 10 mg/kg (day 1 only) | None |

**Randomisation 2: MRD-positive treatment escalation pathway**

**Treatment arm 1 (control): Lenalidomide (R) maintenance (each cycle = 28 days)**

| Treatment (route) | Starting dose (days) |
|---|---|
| Lenalidomide (PO) | 10 mg/kg* (1–21) |

**Treatment arm 2: RBorD consolidation+R maintenance**

| Treatment (route) | Consolidation, starting dose (days) (each cycle = 21 days, 4 cycles) | Maintenance, starting dose (days) (each cycle = 28 days) |
|---|---|---|
| Bortezomib (SC) | 1.3 mg/m$^2$ (1, 8, 15) | NA |
| Lenalidomide (PO) | 15 mg* (1–14) | 10 mg* (1–21) |
| Dexamethasone (PO) | 20 mg (1, 8, 15) | NA |

**Treatment arm 3: RISa maintenance**

| Treatment (route) | Maintenance, starting dose (days) (each cycle = 28 days) | |
|---|---|---|
| Lenalidomide (PO) | 10 mg* (day 1–21) | |
| Isatuximab (IV) | 10 mg/kg (1, 8, 15 and 22 cycle 1 then days 1 and 15 from cycle 2 onwards) | |

**Treatment arm 4: RBorIsaD+RIsa**

| Treatment (route) | Consolidation, starting dose (days) (each cycle = 21 days, 4 cycles) | Maintenance, starting dose (days) (each cycle = 28 days) |
|---|---|---|
| Bortezomib (SC) | 1.3 mg/m$^2$ (1, 8, 15) | NA |
| Lenalidomide (PO) | 15 mg* (1–14) | 10 mg* (1–21) |
| Isatuximab (IV) | 10 mg/kg (1, 8, 15 cycle 1 then days 1 and 8 cycle 2, then days 1 and 15 cycle 3 onwards) | 10 mg/kg (1 and 15) |
| Dexamethasone (PO) | 20 mg (1, 8, 15) | NA |

**Randomisation 3: High risk treatment pathway**

**Treatment arm 1: RBorD consolidation and R maintenance**

| Treatment | Consolidation, starting dose (days) (each cycle = 21 days, 4 cycles) | Maintenance, starting dose (days) (each cycle = 28 days) |
|---|---|---|
| Bortezomib (SC) | 1.3 mg/m$^2$ (1, 8, 15) | NA |
| Lenalidomide (PO) | 15 mg* (1–14) | 10 mg* (1–21) |
| Dexamethasone (PO) | 20 mg (1, 8, 15) | NA |

**Table 2** Continued

| Treatment arm 2: RBorIsaD consolidation+RIsa maintenance | | |
|---|---|---|
| Treatment | Consolidation, starting dose (days) (each cycle = 21 days) | Maintenance, starting dose (days) (each cycle = 28 days) |
| Bortezomib (SC) | 1.3 mg/m² (1, 8, 15) | NA |
| Lenalidomide (PO) | 15 mg* (1–14) | 10 mg* (1–21) |
| Isatuximab (IV) | 10 mg/kg (1, 8, 15 cycle 1 then days 1 and 8 cycle 2, then days 1 and 15 cycle 3 onwards) | 10 mg/kg (1 and 15) |
| Dexamethasone (PO) | 20 mg (1, 8, 15) | NA |

*Or final dose administrated at the end of induction or consolidation treatment if lower.
IV, Intravenous; MRD, minimal residual disease; NA, not applicable; PO, By mouth; SC, Sub-cutaneous.

### Trial follow-up

Participants who discontinue during induction and until 100 days post-ASCT will be followed up for data pertaining to safety, details of salvage treatment, progression (including second progression) and survival at standard care visits. Follow ups will be conducted every 2 months up until disease progression, death or the end of trial. Similar follow-up will be completed for participants who discontinue their allocated trial treatment, which started post-ASCT. In addition consenting participants will still complete QoL and resource use questionnaires at appropriate time points.

All types of adverse event (AEs)/reaction will be collected until 60 days after the last dose of protocol treatment.

Following disease progression, all participants will be followed up annually until death, or until the end of the trial.

### Sample size

In total 1400 participants will be recruited into the trial. Accounting for participants who will be lost to follow-up, this allows for 500, 450 and 172 participants to enter each randomisation, respectively.

### Randomisation 1

Data from our previous trial Myeloma XI, suggests that approximately 50 participants will lose their MRD-negative status during the first 12 cycles of Isa maintenance. In addition, Myeloma XI estimated median PFS with continuous R to be 50 months among TE participants.[32] Under these assumptions and allowing for our randomisation being 12 months later, 500 participants allows for 80% power to test a non-inferiority margin of 10% at 2 years post-R1 (this equates to a HR of 1.38) with a one-sided 5% significance level, allowing for approximately 20% further drop-out after R1 among participants unwilling to cease treatment or being lost to follow-up.

A power of 80% will be attained when 238 PFS events have been observed. These calculations assume that PFS follow an exponential distribution and that there will be 3 years of recruitment to R1 and 2 further years of follow-up (in order that all participants can have PFS at 2 years post-R1 assessed).

Global Health Status/QoL is considered a key secondary endpoint of R1 as the conclusion that the cessation of continuous treatment is non-inferior should not come at the expense of QoL. Ignoring those required to account for participants lost to follow-up, the 406 participants recruited above give >95% power to detect a difference of 10 points at 2 years post-R1, where 10 points represents an accepted medium magnitude clinically relevant difference.[33] The calculation uses data collected from the intensive pathway of MRC Myeloma IX to estimate that the SD in the Global Health Status/QoL subscale of EORTC-QLQ-C30 as 27.54 along with a two-sided 5% significance level.[34]

### Randomisation 2

R2 is a multiarm, multistage drug/drug-combination assessment platform with seamless phase II (activity) and phase III (efficacy) stages. Each experimental arm is compared with a designated control arm (R maintenance).

Early data from Myeloma XI and similar data from IFM/DFCI 2009[35 35] suggest that a 10% upgrade in MRD-negative rate after 6 months continuous R treatment is achievable. It is thought that RBorD consolidation followed by attenuated continuous R or continuous R combined with antibody or immunotherapy could lead to a 30% increase in MRD-negative rate, a 20% upgrade of increase at 6 months post-randomisation when compared with continuous R treatment, as good as therapy combinations containing an alternative proteasome inhibitor.[33]

Following the strategy of Bratton and colleagues[36] for a multiarm multistage trial with:
- ► n=2 stages (Stage 1 (activity) and Stage 2 (efficacy)).
- ► J=4 experimental arms (RBorD+R, R+Isa, RBorIsaD+RIsa and TBC) and a single control arm (R)
- ► Where allocation is equal across each arms (1:1:1:1:1).
- ► The power in stage 1 and stage 2 is 95% and 85%, respectively.
- ► The one-sided significance level in stage 1 and stage 2 is 25% and 1%, respectively.

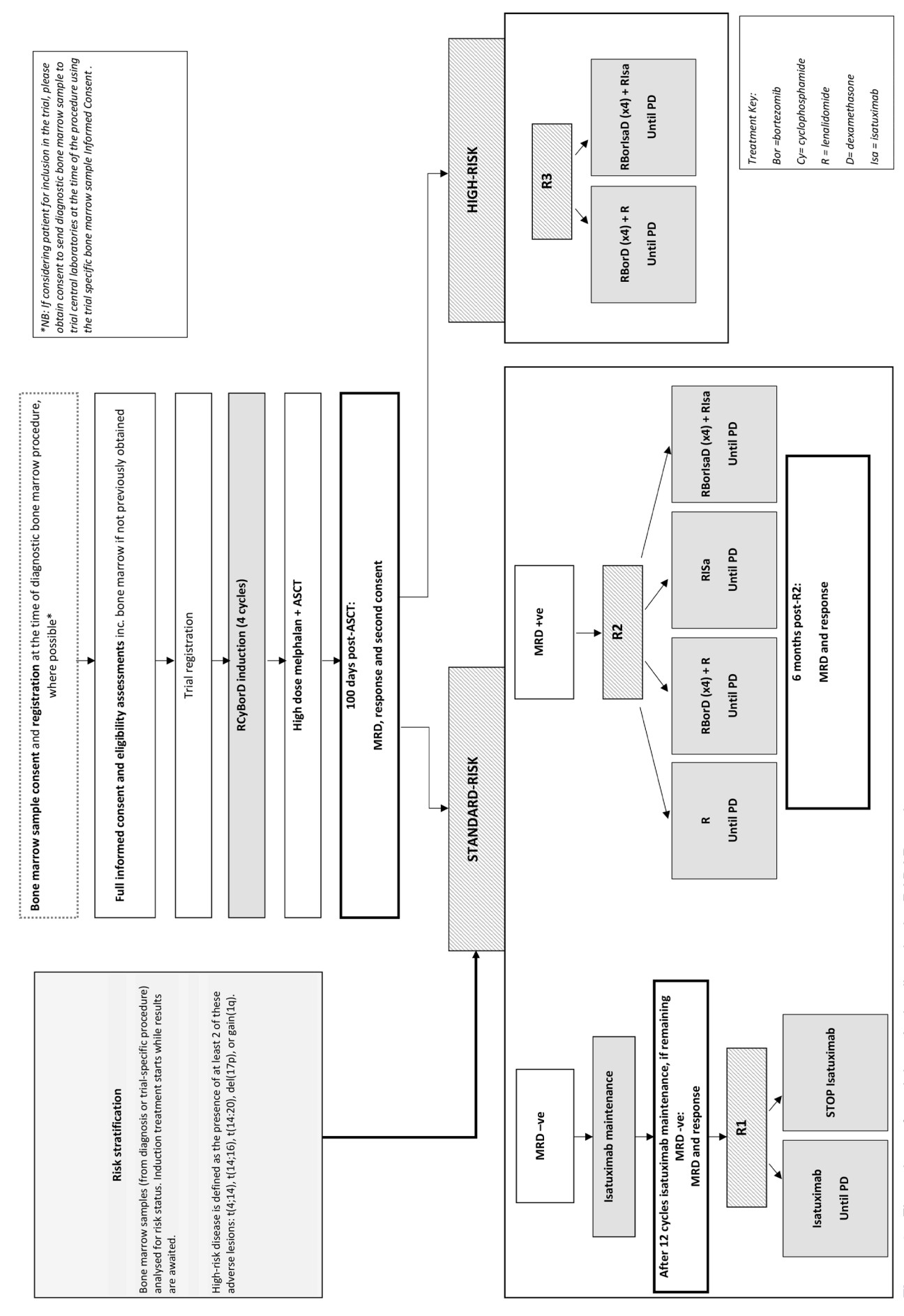

**Figure 1** Flow chart of participant's timelines in the RADAR study.

| Assessment | RCyBorD Induction – All participants | | | | R1 treatment pathway | | | R2 and R3 treatment pathway (R2 participants on R or RIsa arm) | | |
| --- | --- | --- | --- | --- | --- | --- | --- | --- | --- | --- |
| | Pre-registration/baseline (within 14 days prior to registration, unless otherwise stated) | Day 1 (or day -1) of each cycle during RCyBorD induction†† | End of RCyBorD induction | 100 days post-ASCT (within 14 days prior to R2, start of Isatuximab maintenance or R3) | Day 1 of each cycle of Isatuximab treatment†† | On the day of Isatuximab dose†† | Follow up for participants coming off study during R1 treatment pathway: every 2 months | Day 1 of each cycle of treatment†† | On day of (and prior to) every isatuximab dose†† | Follow up for participants coming off study during R2/R3 treatment pathway: every 2 months. After 3,6,12 and 18 cycles** |
| Written informed consent | ✓ Consent to be taken prior to all assessments, permitted outside the 14-day window | | | ✓ (permitted outside the 14-day window) | | | | | | |
| Physical examination | ✓ | ✓ | ✓ | ✓ | ✓ | | | ✓ | | |
| ECOG | ✓ | ✓ | ✓ | ✓ | ✓ | | | ✓ | | |
| Covid vaccination status | ✓ | ✓ | ✓ | ✓ | | | ✓ | ✓ | | ✓ |
| Medical history (including review for prior cancers and history of Covid infection) | ✓ | | ✓ | ✓ | | | | | | |
| Assessment of cardiac risk* | If clinically indicated | | ✓ | If clinically indicated | | If clinically indicated | | | If clinically indicated | |
| ECG | If clinically indicated | If clinically indicated | ✓ | If clinically indicated | Every 3 months in participants receiving Isatuximab | If clinically indicated | | Every 3 months in participants receiving Isatuximab | If clinically indicated | |
| Thyroid function monitoring* | ✓ (within 28 days prior to registration) | ✓ | ✓ | ✓ | ✓ | ✓ FBC, U&E and creatinine only | | ✓ | ✓ FBC, U&E and creatinine only | |
| FBC, LFTs, albumin, LDH, U&Es, calcium, and creatinine¶ | ✓ | ✓ | ✓ | ✓ | ✓ | ✓ FBC, U&E and creatinine only | | ✓ | ✓ FBC, U&E and creatinine only | |
| Hepatitis B, (surface antigen and core antibody), hepatitis C and HIV | ✓ (within 28 days prior to registration) | | | | | | | | | |
| β2-microglobulin (β2M) | ✓ | | ✓ | ✓ | | | | | | |
| C-Reactive protein (CRP) | ✓ | | ✓ | ✓ | | | | | | |
| Calculated creatinine clearance (using Cockcroft-Gault formula) | ✓ | ✓ | ✓ | ✓ | ✓ | | | ✓ | | |
| Serum paraprotein and immunofixation, serum free light chains and urinary free light chain detection | ✓ | ✓ | ✓ | ✓ | ✓ | | ✓ (if performed as part of standard of care) | ✓ | | ✓ (if performed as standard of care) |
| Bone marrow (aspirate and trephine) See Table below for details of central investigations | ✓ (recommended only if a first occurrence of CR or sCR is suspected) | | ✓ | ✓ | ✓ (recommended only if a first occurrence of CR or sCR is suspected) | | ✓ (if performed as part of standard care) | ✓ (recommended only if first occurrence of CR/sCR is suspected) | | ✓ (if performed as standard of care) |
| Pregnancy prevention counselling | ✓ At least every 28 days during lenalidomide treatment | | | | | | | | | |
| Pregnancy test† (for females of childbearing potential) | ✓¥ | ✓¥ | | | ✓ | If clinically indicated | | | ✓¥ | If clinically indicated |
| Cross-sectional imaging according to local practice, to include the thorax (low dose whole body CT recommended) | Within 3 months before registration§ / If clinically indicated.‡ | If clinically indicated.‡ | | | | If clinically indicated | | | If clinically indicated | |
| Assessment of adverse events (including adverse events and second primary malignancy) | Monitor throughout study and report on relevant treatment eCRFs (from first treatment dose until 60 days post last treatment dose) | | | | Monitor throughout study and report on relevant treatment eCRFs (from first treatment dose until 60 days post last treatment dose). AEs should continue to be monitored for participants randomised to 'stop Isatuximab' at R1 (and until 60 days after last on-study assessment). | | | Monitor throughout study and report on relevant treatment eCRFs (from first treatment dose until 60 days post last treatment dose) | | |

*Performed as part of standard of care.

**ECGs should be scheduled to align with other required face-to-face visits

¶The FBC should be repeated mid-cycle 1 (Day 14 +/-3 days) or more frequently and during subsequent cycles if there is a concern about cytopenias

†Females of childbearing potential (FCBP) must have a negative pregnancy test performed by a healthcare professional in accordance with the Celgene Lenalidomide Pregnancy Prevention Plan

††May be performed within the 48 hours prior to the specified day

†† In addition to standard Day 1 assessments

¥All FCBP: The first pregnancy test must be performed within 10 to 14 days prior to the start of lenalidomide and the second pregnancy test must be performed within 24 hours prior to the start of lenalidomide.

FCBP with regular or no menstrual cycles: within 24 hours before start of lenalidomide treatment; weekly for the first 28 days of study participation; 4-weekly as a minimum during lenalidomide treatment; at study discontinuation; and 4 weeks after last dose of lenalidomide

FCBP with irregular menstrual cycles: within 24 hours before start of lenalidomide treatment; weekly for the first 28 days of study participation; 2-weekly as a minimum during lenalidomide treatment; at study discontinuation; and 2 weeks after last dose of lenalidomide

**N.B. During the COVID-19 pandemic, face-to-face assessments must be carried out at least at the beginning of cycles 1 and 2 and at the end of consolidation. Subsequently, participants must be seen by a clinician at least every 3 cycles (and more frequently if clinically indicated or there are concerns).**

**Figure 2** Schedule of local investigations.

► Ten participants per month are entered at R2, stage 1 (activity) analysis and reporting requires 1 month and 5% of participants drop-out.

This equates to 40 participants in each arm at stage I and 50 further participants in each arm at stage 2. This will require a maximum of 46 months of recruitment plus a further 6 months of follow-up for the final participant. Overall, this design has a power of 83.4% and a one-sided family-wise error rate (FWER) of ~3%.

Randomisation 2 has been powered assuming four experimental arms and that all experimental arms will continue to stage 2. In the case that only three experimental arms are implemented or at least one experimental arm is concluded to show a lack of activity after stage 1, the trial will have a higher power than stated above.

### Randomisation 3

Each arm of R3 will take the form of one-stage treatment design proposed by Simon et al[37] where the A'Hern design[38] is first implemented to determine treatment activity in both arms. Treatment activity will be determined by considering the PFS rate at 28 months. Data from Myeloma XI demonstrated a median PFS for high-risk participants in the intensive pathway of approximately 19 months (unpublished data). It is thought that for the treatments proposed to be investigated further in a Phase III trial they should increase the median PFS from 19 months to at least 28 months (9 month increase), that is, increasing the PFS rate at 28 months from 34.7% to 50%.

In order to have an 80% power of demonstrating that the one-sided 95% CI of the PFS rate at 28 months excludes 34.7%, 81 evaluable participants are required, where a treatment arm will be deemed a 'success' if at least 37 out of 81 participants are progression-free at 28 months. To account for approximately 5% of participants being unevaluable at 28 months post-R3, 86 participants will be randomised to each treatment arm that is, 172 participants are required to be entered into R3.

### Total study sample size

Accounting for those participants who will be lost to-follow-up R1, R2 and R3 require 500, 450 and 172 participants to enter each randomisation, respectively.

Considering R1 and R2, it is assumed that 50 participants will lose their MRD-negative status during the 12 months of continuous treatment (Myeloma XI, unpublished data). Therefore, 550 participants are required to enter the MRD-negative part of the standard-risk pathway. It is assumed that 55% of standard-risk participants will be MRD-negative at the point of assessment therefore approximately 1000 participants are required to reach the MRD assessment stage of the standard-risk pathway[35 39] It is assumed that 90% of all those recruited will reach this point therefore 1112 participants are required to enter the standard-risk pathway.[5] It is anticipated that 80% of all those recruited will be standard-risk (Myeloma XI,

unpublished data) therefore at least 1390 participants are required to be recruited to the trial. Rounding this we obtain an overall sample size of 1400.

This gives 280 participants who are expected to enter the high-risk pathway, using the same logic as in the standard-risk pathway if 90% of participants reach R3 then there will be approximately 252 participants recruited to R3. Given the calculations above it is plausible that another arm could be added to the high-risk pathway at a later date.

### Recruitment

It is planned that 1400 participants will be recruited for a total of 36 months. Once all centres are open, the recruitment target is 38 participants a month (equating to approximately 30 standard risk and 8 high-risk participants).

In order to ensure the trial will meet the target sample size within the recruitment period, site set-up was prioritised while the trial was preparing to open to recruitment. In addition, the trial team are actively engaging with PIs at sites who support the trial to ensure that those sites open quickly and are maintaining regular communication with open sites to ensure that they continue to recruit to the trial.

The trial opened to recruitment on 11 May 2021.

### Assignment of treatment allocations

Each of the registration and randomisation procedures will be conducted centrally using the University of Leeds CTRU automated 24-hour web base and telephone system.

### Registration

If a patient is suspected to have myeloma they will enter the trial at the time of routine diagnostic tests, through bone marrow registration. Once diagnosis is confirmed locally and the research team consider the patient potentially eligible for the trial, patients will be provided with full verbal explanation of the trial and full participant information sheet and informed consent documents to consider. Once the patient has provided informed consent to the first stage of the trial they will be registered into the trial at Bone Marrow Registration.

Patients who have already had Myeloma confirmed prior to being approached for the trial, and those who decline consent at bone marrow registration, will enter at main trial registration.

### Data collection

Data collection will primarily be via Remote Data Entry (RDE) on eCRFs, provided by the CTRU at the University of Leeds, and will be entered by staff at the research site. Access to the live RADAR database will be provided by the CTRU following sites being authorised to open to recruitment; guidance on RDE and completing eCRFs will be provided. Data regarding informed consent, QoL, Healthcare Resource Use questionnaires and in-patient stay forms will be collected via paper CRFs.

QoL and Healthcare Resource Use questionnaires will be completed by the participant in clinic or at home and sealed in an envelope before handing back to the site research team who will return them to the CTRU via post.

## Data management

All information collected during the course of the trial will be kept strictly confidential. Information will be held securely on paper and electronically at the CTRU. The CTRU will comply with all aspects of the Data Protection Act 2018.

Data will be monitored for quality and completeness by the CTRU. Missing data will be chased until it is received, confirmed as not available or the trial is at analysis. The CTRU/sponsor will reserve the right to intermittently conduct source data verification exercises on a sample of participants, which will be carried out by staff from the CTRU/Sponsor. Source data verification will involve direct access to participant notes at the participating hospital sites and the ongoing central collection of copies of informed consent documents and other relevant investigation reports.

## Statistical methods

Statistical analysis is the responsibility of the CTRU statisticians, with the exception of the cost-effectiveness analysis, which will be undertaken by the health economists at the University of Leeds. A full statistical analysis plan (SAP) and Health Economics Analysis Plan will be written and approved before any analyses are undertaken.

The analysis population will depend on the randomisation. Separate intention-to-treat (ITT), per-protocol (PP) and safety populations will be defined for each randomisation. In either case the ITT population will include participants according to their randomisation allocation, regardless of eligibility whether they prematurely discontinued the treatment, or did not comply with the regimen. The PP population will exclude individuals who do not receive their randomisation allocation or who are found to be ineligible following randomisation. The safety population will consist of all participants who receive at least one dose of the trial treatments and analyse participants according to what they received rather than what they were randomised to.

With the exception of two interim analyses, no other formal analysis is planned until participants have attained the primary endpoint for each of the randomisations.

### Primary endpoint analyses
#### Randomisation 1

The primary endpoint for R1 will be attained 2 years after the final participant has been randomised to R1.

As R1 is a non-inferiority trial, both PP and ITT population analyses will be used as coprimary analyses and both are required to demonstrate statistical significance of non-inferiority where any discrepancies will be reported.

PFS at R1 will be assessed using Cox proportional hazards regression to adjust for the stratification factors of R1 excluding centre, and presented using Kaplan-Meier estimates of the survival function. The treatment cessation strategy will be declared non-inferior to the continuous treatment strategy if the upper bound of the one-sided 95% CI for the HR is less than 1.38. Note no adjustment will be made for interim analysis due to it being a test for inferiority (see below) and not non-inferiority.

### Randomisation 2

The primary endpoint for R2 will be attained 6 months after the final participant has been randomised to R2. Primary analysis will be undertaken on the ITT population.

An overall one-sided approximately 3% FWER will be applied accounting for each pair-wise comparison over each stage and all comparisons. This equates to a 25% and 1% one sided significance level at interim and final analysis. MRD-negative rate will be compared between each pair-wise comparison (experimental vs control) in turn using logistic regression and adjusted for the minimisation factors of R2 excluding centre.

### Randomisation 3

The primary endpoint for R3 will be attained 28 months after the final participant has been randomised to R3.

Primary analysis will be undertaken on the ITT population. The number of participants who are alive and progression-free at 28 months post R3 will be summarised separately within each treatment arm and exact one sided 95% CIs calculated. The null hypothesis is that the percentage of participants alive and progression-free at 28 months post R3 is 34.7%. A treatment arm is deemed a 'success' if the one sided 95% CIs of the PFS rate at 28 months excludes 34.71%. The treatment will then be proposed to be investigated further in a phase III trial.

No multiple imputation is planned for the primary endpoints.

### Secondary endpoint analyses

OS and other time-to-event endpoints will be analysed using methods similar to those described for PFS. All binary endpoints will be analysed using similar methods to those described for the attainment of MRD negativity endpoint.

For both R1 and R3, each experimental arm will be independently compared with historical control data using molecularly matched individual participant data (IPD) from Myeloma XI+(TE) and assessed for superiority in terms of PFS. Further details of the molecular matching of IPD will be described in the SAP, but it is anticipated that propensity score matching may be an appropriate technique. In each case a one-sided 10% significance level will be applied for each comparison with historical data.

The number and proportion of participants in each response category will be summarised by allocated treatment and exact 95% two-sided confidence intervals will be calculated. The difference in proportions for each

 

response category will be presented with corresponding 95% two-sided confidence intervals.

QoL will be summarised using mean scores adjusted for baseline and two-sided 95% CI for each EORTC QLQ-C30 and EORTC QLQ-MY20 module symptom, role and functioning domain at each assessment time-point. Similar summaries will be produced for quality-adjusted life years (QALYs) as scored by the EQ-5D 3 Level (EQ-5D-3L) questionnaire. For the key secondary analysis of R1, the Global Health Status/QoL subscale at 2 years post-R1, will be compared between the arms using a multivariable linear regression model, adjusting for the R1 stratification factors.

### Exploratory and subgroup analyses

A series of subgroup and exploratory analyses will be undertaken. Subgroup analyses will follow the same structure as the main analysis of the respective endpoint. Where appropriate interaction terms will be added to the regression models to account for the subgroup being investigated. Subgroup and exploratory analyses may, by chance, generate false negative or positive results. Those carried out will be interpreted with caution.

The analysis for the exploratory outcomes will be detailed in the SAP.

### Health economics

The economic evaluation will take a decision modelling approach to assess the cost-effectiveness of treatment options for risk stratified treatment versus standard non-stratified treatment over participants' lifetime. This will allow several research questions to be incorporated within one model. The model will be developed using best practice[40] and the model structure, health states and parameter values will be derived from the trial outputs, published literature, and expert clinical opinion. Data to model stratified treatment options will be drawn from the RADAR trial results while parameters for standard non-stratified treatment will be obtained from the wider literature.

Reflecting the main trial analysis of effectiveness, the economic evaluation will assess the cost-effectiveness of treatment options at each of the randomisations in RADAR. For standard risk participants at R1, the cost-effectiveness analysis will compare treatment with isatuximab for 12 cycles only to long-term isatuximab treatment, continued until disease progression. For standard risk participants who are MRD-positive at 100 days post-ASCT in terms of conversion to MRD negativity at R2, the cost-effectiveness of post-ASCT consolidation and maintenance treatments in the treatment of standard-risk participants will be evaluated. For high-risk participants at R3, the cost-effectiveness of post-ASCT consolidation and maintenance treatment strategies containing lenalidomide, bortezomib, dexamethasone and isatuximab separately will be evaluated. In addition, the cost-effectiveness of stratified treatment pathways will be compared with standard non-stratified treatments.

All analyses will be conducted following National Institute for Health and Care Excellence (NICE) reference case guidelines.[41] As such, outcomes will be measured in QALYs based on EQ-5D[42] and EORTC QLQ-C30[43] utilities and costs will be calculated from a health and personal social services perspective. Cost-effectiveness will be determined in comparison to the NICE cost-effectiveness threshold of £20 000 per QALY gained. Deterministic and probabilistic (using Monte-Carlo simulations) sensitivity analyses will be conducted to explore the impact of parameter value changes on estimates of cost-effectiveness and to determine the level of uncertainty around the results of the lifetime analysis. In addition, value of information analysis will be conducted to estimate the value of further research.

### Trial oversight

The TMG includes the CI, CTRU team and coinvestigators that are assigned responsibility for the clinical set up, ongoing management, promotion of the trial and interpretation of the results. The trial steering committee (TSC), consisting of independent clinicians and statisticians, will provide overall supervision on the trial, including trial progress, adherence to protocol, participant safety and consideration of new information.

### Data monitoring

An independent data monitoring and ethic committee (DMEC) will review the safety and ethics of the trial. Detailed reports will be prepared by the CTRU for the DMEC meetings which will take place at approximately yearly intervals. Additional reports will also be provided at intervals between these meetings. The committee will also review cumulative safety data by arm on an ongoing basis along with individual serious AE (SAE)/SAR listings. After each annual review, the DMEC will make their recommendations to the TSC about the continuation of the trial.

### Interim analyses

Full interim reports will be presented to the DMEC in confidence annually and following the planned interim analyses which are expected to take place when half the number of events (119) have been observed in the treatment de-escalation pathway (R1) and at the end of the activity stage 1 treatment escalation pathway (R2), approximately 6 months after the 40th participant has been randomised.

The interim analysis of R1 will be a one-sided test for inferiority conducted at the 5% significance level. PFS at R1 will be assessed using Cox proportional hazards regression to adjust for the minimisation factors (excluding centre), and presented using Kaplan-Meier estimates of the survival function.

The interim analysis for R2 will be conducted at the end of Stage I (the phase II part) of the randomisation which is due 6 months following the randomisation of the 40th participant in each arm. This analysis will compare

each experimental arm with the control arm in separate tests of superiority using a one-sided 25% significance level and 95% power to determine whether any of the experimental arms show lack of activity, that is, the null hypothesis in the superiority test is not rejected. Should an arm show lack of activity it will not be continued to the second stage of the randomisation (the phase III part). To account for the interim analysis and to give an overall significance level of 1% and power of 80% for each individual comparison the final analysis will be conducted using a one-sided 1% significance level and 85% power. Recruitment will continue seamlessly between activity and efficacy in these scenarios. Formal powered pairwise comparisons are not undertaken comparing experimental arms at either stage.

The DMEC committee members, in light of the interim analysis, will make their recommendations to the TSC about the continuation of the different aspects of the trial.

## Harms
### Adverse events
AEs are any untoward medical occurrence in a participant or clinical trial subject administered a medicinal product and which does not necessarily have a causal relationship with this treatment. AEs can be defined as any unintentional, unfavourable clinical signs or symptoms; any new illness or disease, or the deterioration of existing disease or illness; or clinically relevant deterioration in any laboratory assessments or clinical test. Due to the nature of MM and its treatment, participants are likely to experience several AEs throughout the course of the disease.

All AEs, both related and unrelated to myeloma treatment will be collected on the relevant eCRF from trial registration until 60 days after the last dose of protocol treatment and will be evaluated in accordance with the NCI-CTCAE V.5.0 (NCI-CTCAE). AEs and ARs should continue to be collected for participants who are randomised to 'stop isatuximab' at R1 (and until 60 days after last on-trial assessment), even though they will not be receiving any trial drugs during this period.

### AEs of special interest
An AE of special interest (serious or non-serious) is one of scientific and medical concern specific to isatuximab, for which ongoing monitoring and rapid communication by the investigator to the sponsor is considered to be appropriate. Isatuximab AEs of special interest include; ≥grade 3 infusion reactions (IRs), pregnancies, symptomatic overdoses with IMP and second primary malignancies (SPMs).

### Serious AEs
SAEs are defined as any untoward medical occurrence or effect that at any dose results in death; or are life threatening (at the time of the event); or require in-patient hospitalisation or prolongation of existing hospitalisation; or result in persistent or significant disability or incapacity; or result in a congenital anomaly or birth defect; or any other important medical event. All SAEs will be reported from registration until 60 days post the last dose of the trial drug. SAEs should continue to be collected for participants who are randomised to 'stop isatuximab' at R1 (and until 60 days after last on-study assessment), even though they will not be receiving any study drugs during this period.

Serious adverse reactions (SARs) are SAEs that are deemed possibly related to any dose administered of any trial treatment. Suspected unexpected serious adverse events (SUSARs) are SARs which are not listed in the reference safety information for that medicinal product. SARs and SUSARs will be reported from the date of first trial dose and for the duration of the trial.

### Secondary malignancies
All new primary malignancies (SPMs) or suspected malignancies occurring from the date of trial registration and for the duration of the trial will be recorded. SPMs will be summarised and reviewed by an appointed member of the TMG who will determine whether trial treatment should continue.

### Pregnancies
Participants registered into RADAR agree to follow the Celgene approved Pregnancy Prevention Programme. Pregnancies and suspected pregnancies (including a positive pregnancy test regardless of age or disease status) in a female participant must be reported throughout the study and for 12 months after the last dose of cyclophosphamide or for 5 months following cessation of protocol treatment, whichever is longest. Pregnancies and suspected pregnancies (including a positive pregnancy test regardless of age or disease status) in a male participant's partner must be reported throughout the study and for 6 months after the last dose of cyclophosphamide or for 5 months following cessation of protocol treatment, whichever is longest.

### Safety analyses
Safety analyses will summarise all SUSARs, SARs, SAEs, AEs and ARs, respective to where they occurred within the trial pathway (Induction, ASCT, Pre-R1, Post-R1, Post-R2, Post-R3). Safety data will be presented by treatment group for the safety population in addition to relationship to trial treatment.

### Auditing
The CTRU and the trial sponsor have procedures in place to ensure that serious breaches of Good Clinical Practice (GCP) or the trial protocol are picked up and reported. A triggered monitoring plan will ensure that sites at risk are monitored accordingly.

## Patient and public involvement
RADAR has been developed following extensive discussion with the UK myeloma community. Patients were first involved into the research at the grant application

stage, and were incorporated into the development of the primary endpoint of R1, in which they confirmed a research question involving the stopping of treatment given there was no detectable myeloma would be of interest. The trial consent and participant information documents were reviewed for clarity by a patient representative. To ensure that a patient perspective is considered throughout the trial a PPI advisory group, including 4–8 members will feed into TMG meetings, and plan to host quarterly meetings to discuss urgent matters. It is envisioned that the PPI group, may help with writing summaries of results that can be published on patient forums and websites and maybe even shared directly with patients.

## Ethics and dissemination
### Research and ethics approval
Ethics approval has been obtained from the London–Central Research Ethics Committee (20/LO/0238). In addition capacity and capability has been confirmed by the appropriate research and development department for each participating centre prior to opening to participant recruitment into the trial. This included but was not limited to; having access to the non-trial supplied IMP (Bor, Cy and D), being able to conduct FISH as SoC, capacity of staff, ability to complete all protocol required assessments/investigations, and ability to archive trials documents.

Participants will be required to provide informed consent before joining the trial.

### Protocol amendments
The trial opened to recruitment on 11 May 2021 using protocol version 2, dated 29 April 2020. This publication is written based from protocol version 3 dated July 2021. An amendment to protocol version 3 is anticipated Mid-2022, which should include the following:

#### *High-risk pathway*
The cytogenetic abnormality del(1p) will be added to the definition of high risk. In addition, Isa will be added to induction regimen for high-risk patients following their first cycle of RCyBorD (while their risk status is confirmed). Furthermore, R3 will be removed and high-risk patients will receive RBorIsaD+RIsa only post-ASCT. The pathway will follow a single-arm, single stage, phase II, three outcome design, with the new primary objective to assess the activity of RCyBorD+Isa induction followed by ASCT and RBorD+Isa consolidation and RIsa maintenance in terms of PFS at 18 months post-trial registration.

#### *Other changes*
► The removal of the phone registration and randomisation.
► The clarification of allowing prescribed medication to remove risks of thromboembolism before preceding onto treatment in the exclusion criteria for registration (and for starting post ASCT treatment).

► The allowance of blood support products, blood transfusions and platelet transfusions as per institutional guidelines at eligibility and pretreatment assessments.
► An update of exclusion criteria to allow for previous plasmacytomas.

### Consent or assent
The PI retains overall responsibility for the informed consent of participants at their site and must ensure that any medically qualified person delegated responsibility to participate in the informed consent process is duly authorised, trained and competent to participate according to the ethically approved protocol, principles of GCP and Declaration of Helsinki. Written consent will be obtained and signed by a medically qualified member of the site research team. Informed consent must be obtained, and the participant must be registered into the trial prior to the participant undergoing procedures that are specifically for the purposes of the trial and are out-with standard routine care at the participating site. This include the confirmation of optional consent (QoL/healthcare resource use questionnaires and decision regarding consent for samples to be used in future research). At any stage, participants can withdraw consent without repercussion.

### Confidentiality
All information collected during the course of the trial will be kept strictly confidential. Information will be held securely on paper and electronically at the CTRU. The CTRU will comply with all aspects of the Data Protection Act 2018.

### Access to data
IPD (with any relevant supporting material, for example, data dictionary, protocol, SAP) for all trial participants (excluding any trial-specific participant opt-outs) will be made available for secondary research purposes at the end of the trial, that is, usually when all primary and secondary endpoints have been met and all key analyses are complete.

Data will be shared according to a controlled access approach, based on the following principles:
► The value of the proposal will be considered in terms of the strategic priorities of the CTRU, chief investigator and sponsor, the scientific value of the proposed project, and the resources necessary and available to satisfy any data release request.
► We encourage a collaborative approach to data sharing, and believe it is best practice for researchers who generated datasets to be involved in subsequent uses of those datasets.
► The timing and nature of any data release must not adversely interfere with the integrity of the trial or research project objectives, including any associated secondary and exploratory research objectives detailed in the ethically approved original research protocol. On an individual trial or research project basis, a reasonable period of exclusivity will be agreed with the trial or research project team.

| Sample | Investigation | Diagnosis (if patient consented on NHS form and trial bone marrow Informed Consent Document)[1] | Baseline (post trial consent and before the start of treatment) | Day 1 of Cycle 1 RCyBorD induction | End of induction cycle 1 | End of induction cycle 2 | End of RCyBorD induction | 100 days post-ASCT and prior to consolidation/maintenance |
|---|---|---|---|---|---|---|---|---|
| 0.5-1 mL EDTA bone marrow aspirate **SEND TO HMDS, LEEDS** | Minimal Residual Disease | ✓ | ✓ (if not sent previously) | | | | ✓ | ✓ |
| 10 mL clotted peripheral blood / Random urine sample **SEND TO BIRMINGHAM** | Disease parameters (including paraproteins) for response assessments and covid-19 antibody response | | ✓ | ✓ | ✓ | ✓ | ✓ | ✓ |
| | M-protein intereference assay | | | Only if baseline sample not within 14 days | | | | |
| 10 mL EDTA bone marrow aspirate / 10 mL blood **SEND TO UCL, LONDON** | Tumour DNA for neoantigens / cfDNA / Immune profiling | ✓ Bone marrow only | ✓ BMA (if not sent previously) & blood | ✓ Cycle 1 only — 10 mL blood (only if baseline sample is not from within 14 days prior to cycle 1 day 1) | | | | ✓ |
| Bone marrow trephine (6 sections) **SEND TO UCL, LONDON** | Immunostaining for immune function | ✓ | ✓ (if not sent previously) | | | | | ✓ |

Trial consent — Consented to Myeloma XV trial and central laboratory investigations

[1] Baseline bone marrow samples may be taken before trial consent, provided that patients have consented to these samples being taken using the standard NHS consent form. The patient must have provided written informed trial consent on the trial bone marrow Informed Consent Document before these samples can be sent to the central laboratories.

**Figure 3** Central investigations from diagnosis up to 100 days post-ASCT. ASCT, autologus stem cell transplant.

| Sample | Investigation | Standard-risk MRD-negative participants (R1 pathway) | Standard-risk MRD-positive participants on R or RIsa arm (R2 pathway) | | | Standard-risk MRD-positive participants on RBorD+R or RBorIsaD+RIsa arm (R2 pathway) | | | High-risk participants (R3 pathway) | | | All participants |
|---|---|---|---|---|---|---|---|---|---|---|---|---|
| | | After cycles 6, 12 and 24 of maintenance (or at equivalent time-points if randomised to stop receiving Isa) | After cycle 3 | After cycles 6, 12 and 18 | After cycle 24 | After consolidation cycle 4 | After maintenance cycles 3, 9 and 15 | After maintenance cycle 21 | After consolidation cycle 4 | After maintenance cycles 3, 9 and 15 | After maintenance cycle 21 | Disease progression |
| **0.5-1 mL EDTA bone marrow aspirate** SEND TO HMDS, LEEDS | Minimal Residual Disease | ✓ | ✓ | ✓ | | ✓ | ✓ | | ✓ | ✓ | | ✓ |
| **10 mL clotted peripheral blood** **Random urine sample** SEND TO BIRMINGHAM | Disease parameters (including paraproteins) for response assessments and covid-19 antibody response | ✓ | ✓ | ✓ | ✓ (blood only) | ✓ | ✓ | ✓ (blood only) | ✓ | ✓ | ✓ (blood only) | ✓ |
| | M-protein interference assay | | | | | | | | | | | |
| **10 mL EDTA bone marrow aspirate** **10 mL blood** SEND TO UCL, LONDON | Tumour DNA for neoantigens | ✓ (only after 6 cycles) | ✓ | | | ✓ | | | ✓ | | | ✓ |
| | cfDNA | | | | | | | | | | | |
| | Immune profiling | | | | | | | | | | | |
| **Bone marrow trephine** SEND TO UCL, LONDON | Immuno-chemistry for immune function | | ✓ | | | ✓ | | | ✓ | | | ✓ |

**Figure 4** Central investigations for the R1, R2 and R3 pathways.

► Any data release must be lawful, in line with participants' rights and must not compromise patient confidentiality. Where the purposes of the project can be achieved by using anonymised or aggregate data this will always be used. We will release individual patient data only in a form adjusted so that recipients of the data cannot identify individual participants by any reasonably likely means. We will also only share data when there is a binding agreement in place stating that data recipients will not attempt to re-identify any individual participants.

► Any data release must be in line with any contractual obligations to which the CTRU is subject.

► The research must be carried out by a bone fide researcher with the necessary skills and resources to conduct the research project.

► The research project must have clear objectives and use appropriate research methods.

► The research must be carried out on behalf of a reputable organisation that can demonstrate appropriate IT security standards to ensure the data are protected and to minimise the risk of unauthorised disclosure.

Data will only be shared for participants who have given consent to use of their data for secondary research.

Requests to access trial data should be made to CTRU-DataAccess@leeds.ac.uk in the first instance. Requests will be reviewed (based on the above principles) by relevant stakeholders. No data will be released before an appropriate agreement is in place setting out the conditions of release. The agreement will govern data retention requirements, which will usually stipulate that data recipients must delete their copy of the data at the end of the planned project.

### Ancillary and post-trial agreements
Participants who stop trial treatment due to progression or any point prior to the end of trial will be treated off-trial at the discretion of their treating clinician. Those who stop prior to disease progression will be followed up 2 monthly until disease progression, death or the end of the trial. Following disease progression all participants will be followed up annually until death, or until the end of the trial for post progression endpoints.

Participants in the R1 pathway who are MRD-positive at 12 months post 12 cycles of isatuximab and therefore aren't eligible for randomisation 1 will continue to receive isatuximab maintenance, these participants would continue to be monitored, and data collected, as previously but trial samples would not be collected.

### Dissemination policy
Authorship will be in keeping with the UK-MRA publication policy and due acknowledgement to participants, local investigators, funders and NCRI Haematological Oncology Study Group support made. The success of the trial depends on the collaboration of all participants. For this reason, credit for the main results will be given to all those who have collaborated in the trial, through

authorship and contributor-ship. Uniform requirements for authorship for manuscripts submitted to medical journals will guide authorship decisions alongside the guidance of the UK Myeloma Research Alliance Authorship Policy.

To maintain the scientific integrity of the trial, data will not be released prior to the end of the trial or a primary endpoint being reached, either for trial publication or oral presentation purposes, without the permission of the TSC and the (co-)chief investigators. In addition, individual collaborators must not publish data concerning their participants that is directly relevant to the questions posed in the trial until the main results of the trial have been published and following written consent from the Sponsor.

### Appendices
#### *Informed consent material*
The consent forms which are to be completed by the participant at Bone Marrow Registration and/or Trial Registration and prior to each randomisation pathways are included in online supplemental material.

#### *Biological specimens*
The collection of central samples for laboratory analysis is summarised in figures 3 and 4. The analysis to be conducted for trial purposes is stated in online supplemental material. Additional analysis may be carried out by each central laboratory provided the appropriate consent for sample use in future research has been provided by the participant at trial entry.

**Author affiliations**
[1]Leeds Cancer Research UK Clinical Trials Unit, Leeds Insitute of Clinical Trials Research, University of Leeds, Leeds, UK
[2]Radcliffe Department of Medicine, Oxford University Hospitals NHS Foundation Trust, Oxford, UK
[3]Department of Haematology, UCL Cancer Institute, London, UK
[4]Clinical Immunology Service, Institute of Immunology and Immunotherapy, Medical School, University of Birmingham, Birmingham, UK
[5]Centre for Myeloma Research, Division of Molecular Pathology, Institute of Cancer Research, London, UK
[6]HMDS, St James's University Hospital, Leeds, UK
[7]Department of Immunology and Inflammation, Imperial College London, London, UK
[8]Langmuir Centre for Myeloma Research, Imperial College London, London, UK
[9]Leeds Cancer Centre, St James's University Hospital, Leeds, UK
[10]Academic Unit of Health Economics, Leeds Institute of Health Sciences, University of Leeds, Leeds, UK
[11]Department of Haematology, University of Cambridge, Cambridge, UK
[12]Department of Haematology, Queen Elizabeth Hospital, Birmingham, UK
[13]Department of Haematology, University College London Hospitals NHS Foundation Trust, London, UK
[14]Northern Centre for Cancer Care, Freeman Hospital, Newcastle-Upon-Tyne, UK
[15]Department of Haematology, University Hospital of Wales, Cardiff, UK
[16]Department of Haematology, Royal Hallamshire Hospital, Sheffield, UK
[17]Department of Haematology, Leeds Teaching Hospitals NHS Trust, Leeds, UK
[18]Institute of Cancer and Genomic Sciences, University of Birmingham, Birmingham, UK
[19]Bristol Myers Squibb, Boundry, Switzerland

**Contributors** The RADAR trial was conceived by KY and MC and designed by them in collaboration with DAC. All authors (K-LR, ABC, KR, DAC, AH, SQ, MD, MK,

RO, HWA, GC, DM, CO, LB, RL, AP, BD, MC, GP, RP, GJ, CB, JS, RdT, AC, CP, MC, SA and KY) inputted into the development of the protocol and participant information sheet. The first draft of the protocol paper was written by K-LR and ABC. All authors (K-LR, ABC, KR, DAC, AH, SQ, MD, MK, RO, HWA, GC, DM, CO, LB, RL, AP, BD, MC, GP, RP, GJ, CB, JS, RdT, AC, CP, MC, SA and KY) reviewed and approved the final manuscript.

**Funding** This trial is funded by Cancer Research UK (C9203/A24078), Sanofi Genzyme and Celgene: A BMS Company. This work was also supported by Core Clinical Trials Unit Infrastructure from Cancer Research UK (C7852/A25447). The trial sponsor is responsible for the overall conduct of the trial as defined by Directive 2001/20/EC is University of Leeds, UoL/LTHT Joint Sponsor QA office (CTIMPs); Research & Innovation Centre/Faculty of Medicine & Health; Leeds Teaching Hospitals NHS Trust/University of Leeds; St James University Hospital; Leeds LS9 7TF.

**Disclaimer** The funders had no role in the design, collection, analysis or collection of data; in writing the manuscript; or in the decision to submit the manuscript for publication.

**Competing interests** ABC, KLR, DAC, AH, CO, LBai, LBar, AP and RL report grants and non-financial support from BMS/Celgene, grants and non-financial support from Merck Sharpe & Dohme, grants and non-financial support from Amgen, grants and non-financial support from Takeda, during the conduct of the trial. DAC also reports travel support from Celgene Corporation. KY, RDT, CB, GJ, GP, MC, BD, DM, GC, HA, KR, SA and AC have no conflicting interests to declare. MC declares, Bristol Myers Squibb- employee, Honoraria/travel support in the last 3 years from, Amgen, BMS/Celgene, Janssen, Takeda, Abbvie. CP declares BMS/Celgene Ad boards and speaker fees, Sanotif—Ad board, speaker fees, conference registration fees. JS reports Carrying out consultancy work (Advisory Board) for Sanofi. And an educational speaking engagement for Celgene/BMS RP declares; Honoraria—Jannsen, BMS, Abbvie, GSK. Consultancy: GSK, Janssen. Meeting support: Janssen, Takeda, BMS. RO declares- Janssen - advisory board, honoraria, Celegene—honoraria, Beigene - advisory board, honoraria, Astra Zeneca—honoraria. MK declares inter-relationships: AbbVie: consultancy; Amgen: honoraria; BMS/Celgene: consultancy, research funding (institution); GSK: consultancy; Janssen: consultancy, research funding (institution); Karyopharm: consultancy; Pfizer: consultancy; SeattleGenetics: consultancy; Takeda: consultancy; Sanofi: honoraria. MD reports owning stock in Abingdon Health. SQ is the founder and CSO of Achilles therapeutics a company developing T cell therapies for solid tumours.

**Patient and public involvement** Patients and/or the public were involved in the design, or conduct, or reporting, or dissemination plans of this research. Refer to the Methods section for further details.

**Patient consent for publication** Not applicable.

**Provenance and peer review** Not commissioned; externally peer reviewed.

**Open access** This is an open access article distributed in accordance with the Creative Commons Attribution 4.0 Unported (CC BY 4.0) license, which permits others to copy, redistribute, remix, transform and build upon this work for any purpose, provided the original work is properly cited, a link to the licence is given, and indication of whether changes were made. See: https://creativecommons.org/licenses/by/4.0/.

**ORCID iDs**
Kara-Louise Royle http://orcid.org/0000-0003-0225-1199
Amy Beth Coulson http://orcid.org/0000-0002-1810-409X
Holger W Auner http://orcid.org/0000-0003-4040-0642

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
