## [Reviewer comments · BMJ Open]

ARTICLE DETAILS

TITLE (PROVISIONAL)	Risk and response adapted therapy following autologous stem cell transplant in patients with newly diagnosed multiple myeloma (RADAR (UK-MRA Myeloma XV Trial): study protocol for a Phase II/III randomised controlled trial
AUTHORS	Royle, Kara-Louise; Coulson, Amy; Ramasamy, Karthik; Cairns, David; Hockaday, Anna; Quezada, Sergio; Drayson, Mark; Kaiser, Martin; Owen, Roger; Auner, Holger; Cook, Gordon; Meads, David; Olivier, Catherine; Barnard, Lorna; Lambkin, Rhiannon; Paterson, Andrea; Dawkins, Bryony; Chapman, Mike; Pratt, Guy; Popat, Rakesh; Jackson, Graham; Bygrave, Ceri; Sive, Jonathan; de Tute, Ruth; Chantry, Andrew; Parrish, Christopher; Cook, Mark; Asher, Samir; Yong, Kwee

VERSION 1 – REVIEW

REVIEWER	Wu, Depei Soochow University, the First Affiliated Hospital of Soochow University
REVIEW RETURNED	18-Apr-2022

GENERAL COMMENTS	This RCT study aims to find the suitable therapeutic pathways for NDMM TE patients post-ASCT according to the risk levels and MRD status. There are some questions: 1. Can MM patients with amyloidosis be included in the study?2. For the first pathway, during isatuximab maintenance, if MRD turns positive once and then negative again, can such patients continue with R1?3. Line 25 of Page 14, "Participants identified as standard-risk and who are MRD-positive post-ASCT and have at least a minimal response (MR) at 100 days post ASCT", whether "MR" was correct? While on the line 44 of Page 4, "Patients achieving <PR to induction therapy will not be able to proceed with on study ASCT and will have to be treated off -trial". Please confirm the consistency of the two statements.4. For the second pathway, some standard-risk patients with MRD +ve receive only R until PD, the other three groups were given more intensive treatment regimens. This seems to have an effect on survival outcomes. Please explain it.5. Line 27 of Page 18, "Original/copy of the FISH testing report will also be sent to CTRU where they will be reviewed by the CI/Trial management team for risk stratification". Should it be more appropriate to take the original specimen for confirmation, rather than just the confirmation report?6. Page 39, Bone marrow aspiration was done every two months, isn't it a little frequent?
--

REVIEWER	Kumar, Dr lalit AIIMS Delhi, Medical Oncology
REVIEW RETURNED	21-Apr-2022

GENERAL COMMENTS	This is a risk adapted, multicentric study to evaluate de-escalation of therapy in myeloma patients who are MRD negative after stem cell transplant and also to escalate therapy in patients who are MRD positive or have high risk cytogenetics. Study involves 1400 patients of newly diagnosed MM to be treated in 70 centres in UK over 3 years. Protocol is well written. Following points may be addressed for better clarification.  1. Among eligibility criterias  a. Age >18 years , but upper limit has not been defined. Since ASCT is being planned- tolerance and outcome may vary for patients <60 Y, 60-70/75 and those above 75 Y and above. b. GFR <30 ml/mt . It is suggested that cut off of <40 ml may be considered as per international literature. This may be calculated by MDRD formula which is better than Cockcroft-Gault formula. c. Imaging : imaging has not been specified- is it routine x-ray survey, what about in those with normal skeletal survey- are MRI or low dose CT scan or PET scan being planned? d. Hb ≥8 G/dl has been taken as cutoff, reasons are not clear, what about patients who have low Hb, as getting blood transfusion in preceding days will exclude them. e. Similarly platelet counts cutoff is 75000/cmm. This may also be clarified. f. Cytogenetics- 2 or more high risk cytogenetic abnormality will be considered as high risk . Generally having one of those specified in itself is considered high risk e.g. del p17 alone is high risk. Please explain. 2. Induction therapy  a. Dose of lenalidomide is 25 mg/days x 14 days every 28 days, this is lower than standard of 21 days. Further a number of patients may have eGFR on lower side. b. Dose of dexamethasone is also 40 mg/week x 3 doses as against standard practice of 4 doses. c. Cyclophosphamide- it has been mentioned that cyclophosphamide will be given orally , as per schedule mentioned dose is 500 mg/m² x 2 doses. Would that not be difficult? Do investigators plan to split and give over 21 or 28 days. 3. MRD  a. Is MRD being planned one time day +100 ? or pre and post transplant or at 6 months and 12 months and 24 months. This will help to know impact of persistent MRD negativity or intermittent positivity on the outcome. 4. Treatment  a. Treatment plan for patients who have suboptimal response to induction therapy (<PR) or are found transplant ineligible is not clear. b. Similarly patients who relapse or progress after induction therapy and for those who progress after ASCT may be given.
--

VERSION 1 – AUTHOR RESPONSE

Reviewer 1:

This RCT study aims to find the suitable therapeutic pathways for NDMM TE patients post-ASCT according to the risk levels and MRD status. There are some questions:

1. Can MM patients with amyloidosis be included in the study?

Yes, as long as they also are defined to have multiple myeloma. Only patients with Asymptomatic (smouldering) MM, monoclonal gammopathy of undetermined significance (MGUS), solitary plasmacytoma of bone, or extramedullary plasmacytoma (without evidence of MM) are excluded in this context as per Table 1 of the manuscript.

2. For the first pathway, during isatuximab maintenance, if MRD turns positive once and then negative again, can such patients continue with R1?

No, R1 requires participants to remain MRD-Negative for the whole 12 months.

3. Line 25 of Page 14, "Participants identified as standard-risk and who are MRD-positive post-ASCT and have at least a minimal response (MR) at 100 days post ASCT", whether "MR" was correct? While on the line 44 of Page 4, " Patients achieving

The two sentences refer to the response requirements at two different stages in the trial. Prior to ASCT participants are required to have at least a PR. However, after ASCT participants only require an MR to proceed to post-ASCT treatments.

4. For the second pathway, some standard-risk patients with MRD +ve receive only R until PD, the other three groups were given more intensive treatment regimens. This seems to have an effect on survival outcomes. Please explain it.

R until progression is the control treatment for the R2 comparisons. Current standard of care in the UK is lenalidomide until progression. The trial is investigating whether more intensive treatments provide better outcomes for patients.

5. Line 27 of Page 18, "Original/copy of the FISH testing report will also be sent to CTRU where they will be reviewed by the CI/Trial management team for risk stratification". Should it be more appropriate to take the original specimen for confirmation, rather than just the confirmation report?

FISH testing is conducted as part of routine care in the UK. One of the aims of the trial is to see whether FISH results can be used to determine treatment up front. If a centralised laboratory conducted this testing rather than local laboratories the trial would not be able to show that local FISH testing, conducted as part of routine clinical care for newly diagnosed transplant eligible myeloma patients across different sites, can be used to risk stratify patients effectively without going through a central laboratory. The trial would also not be able to identify whether the local laboratory standards in the UK for myeloma FISH testing are adequate or if there are improvements that can be made in line with national guidance.

6. Page 39, Bone marrow aspiration was done every two months, isn't it a little frequent?

Figure 2 refers to the schedule of local investigations. The 2-monthly bone marrow samples are only to be performed if done as part of standard of care. Mandated bone marrows are less frequent and relate specifically to the key response and endpoint assessments. The mandated bone marrows are detailed in Figures 3 and 4.

Reviewer 2:

This is a risk adapted, multicentric study to evaluate de-escalation of therapy in myeloma patients who are MRD negative after stem cell transplant and also to escalate therapy in patients who are MRD positive or have high risk cytogenetics. Study involves 1400 patients of newly diagnosed MM to be treated in 70 centres in UK over 3 years. Protocol is well written. Following points may be addressed for better clarification.

1. Among eligibility criterias

a. Age >18 years, but upper limit has not been defined. Since ASCT is being planned- tolerance and outcome may vary for patients <60 Y, 60-70/75 and those above 75 Y and above.

The UK does not specify an upper age limit for ASCT, and age alone does not determine whether a patient is transplant eligible in the UK. Instead, an overall assessment of the patient's "fitness" is used by each local site and age will therefore differ from patient to patient. However, age will be included as a subgroup during the planned subgroup analysis of the primary endpoints of the trial, to see whether there is an interaction between age and outcome. The planned subgroup analysis is detailed within the supplementary material.

b. GFR <30 ml/mt. It is suggested that cut off of <40 ml may be considered as per international literature. This may be calculated by MDRD formula which is better than Cockcroft-Gault formula.

In the UK an eGFR < 30 ml / min is defined as clinically concerning kidney disease (Mayo Clinic Stage 3b) and there are dose recommendations for each of the trial IMPs using this cut-off. We have therefore incorporated this definition into our trial protocol.

c. Imaging: imaging has not been specified- is it routine x-ray survey, what about in those with normal skeletal survey- are MRI or low dose CT scan or PET scan being planned?

Imaging is required to be cross-sectional as per local practice throughout the trial although a low dose whole body CT is recommended. X-ray surveys are not recommended as a diagnostic imaging modality for myeloma. The imaging assessments for the trial are detailed in the schedule of local investigations in Figure 2.

d. Hb \geq 8 G/dl has been taken as cutoff, reasons are not clear, what about patients who have low Hb, as getting blood transfusion in preceding days will exclude them.

As the trial is based in the UK, UK standard practice of Hb \geq 80g/L has been incorporated into the protocol to ensure that chemotherapy can be administered safely. However, when the next protocol amendment is implemented blood product support and transfusions will be allowed according to recruiting sites institutional guidelines prior to trial entry and the start of each treatment cycle. This information is now detailed within the future amendments section.

e. Similarly, platelet counts cutoff is 75000/cmm. This may also be clarified.

Similar to above, as the trial is based in the UK, UK standard practice of platelets \geq 75 \times 10⁹/L has been incorporated into the protocol to ensure that chemotherapy can be administered safely. However, when the next protocol amendment is implemented platelet transfusions will be allowed according to recruiting sites institutional guidelines prior to trial entry and the start of each treatment cycle. This information is now detailed within the future amendments section.

f. Cytogenetics- 2 or more high risk cytogenetic abnormality will be considered as high risk. Generally having one of those specified in itself is considered high risk e.g., del p17 alone is high risk. Please explain.

With respect, the definition of “true” high risk has evolved with improved induction regimens, so that single high-risk features today no longer reliably identify a true high-risk group, while the co-occurrence of 2 or more high risk features continues to impart inferior outcomes. This is the current perspective of the UK Myeloma community and others globally:
<https://pubmed.ncbi.nlm.nih.gov/34898239/>

2. Induction therapy

a. Dose of lenalidomide is 25 mg/days x 14 days every 28 days, this is lower than standard of 21 days. Further a number of patients may have eGFR on lower side.

As per Table 7 of the manuscript for the 21-day induction and consolidation cycles lenalidomide is given for 14 days. For the 28-day cycles lenalidomide is given for 21 days. This is in line with standard practice. Only the starting doses are listed in Table 7. Patients are permitted to have their dose modified upfront in the event of renal impairment. This has been clarified in the dose modifications and discontinuations section of the manuscript.

b. Dose of dexamethasone is also 40 mg/week x 3 doses as against standard practice of 4 doses.

Standard practice is weekly dexamethasone. As the induction and consolidation cycles are 21 days (Table 7), 3 doses is appropriate.

c. Cyclophosphamide- it has been mentioned that cyclophosphamide will be given orally, as per schedule mentioned dose is 500 mg/m² x 2 doses. Would that not be difficult? Do investigators plan to split and give over 21 or 28 days.

As per Table 7 of the manuscript a flat dose of 500mg Cyclophosphamide is taken orally on day 1 and 8 of the 21-day induction cycles.

3. MRD

a. Is MRD being planned one time day +100? or pre- and post-transplant or at 6 months and 12 months and 24 months. This will help to know impact of persistent MRD negativity or intermittent positivity on the outcome.

The MRD assessments are at baseline, end of induction and 100 days post-ASCT for all participants. The remaining assessments are dependent on the pathway participants follow post-ASCT and are aligned for consistency across the pathways. For those standard-risk and MRD negative at 100 days post-ASCT their further assessments will be after 6,12 and 24 maintenance treatment cycles. For those who are assigned consolidation and maintenance combination treatment MRD assessments are at the end of consolidation and after 3,9 and 15 maintenance cycles. For those who are assigned maintenance treatment only MRD assessments are after cycle 3,6,12 and 18. This will enable the 6 months post-ASCT treatment to be assessed for every pathway as well as sustained MRD negativity between 100 days post-ASCT and 12 months post-ASCT treatment as well as between 6 and 18 months post-ASCT or 12 and 24 if on the R1 pathway. These assessments are detailed in Figure 3 and Figure 4 and the analysis falls under the “duration of MRD-negativity” secondary endpoint which was missed from the original manuscript.

4. Treatment

a. Treatment plan for patients who have suboptimal response to induction therapy (

Participants who have a suboptimal response to induction therapy are to be treated off-trial as detailed in the limitations section at the start of the manuscript. Participants who do not receive any part of the ASCT + HDM treatment do not proceed to the post-ASCT treatment part of the trial as detailed in the inclusion and exclusion criteria of the trial (Table 2 – 5).

b. Similarly, patients who relapse or progress after induction therapy and for those who progress after ASCT may be given.

Additional clarification has been added to the post-ASCT section of the participant timelines section of the manuscript. Post-ASCT treatment is given until progression, withdrawal or death after which point treatment is as per local practice.

VERSION 2 – REVIEW

REVIEWER	Wu, Depei Soochow University, the First Affiliated Hospital of Soochow University
REVIEW RETURNED	15-Aug-2022
GENERAL COMMENTS	All the questions were well answered
REVIEWER	Kumar, Dr lalit AIIMS Delhi, Medical Oncology
REVIEW RETURNED	03-Aug-2022
GENERAL COMMENTS	None